# Optical framed knots as information carriers

Hugo Larocque[1,2✉], Alessio D'Errico[1], Manuel F. Ferrer-Garcia[1], Avishy Carmi[3], Eliahu Cohen [4] & Ebrahim Karimi [1✉]

Modern beam shaping techniques have enabled the generation of optical fields displaying a wealth of structural features, which include three-dimensional topologies such as Möbius, ribbon strips and knots. However, unlike simpler types of structured light, the topological properties of these optical fields have hitherto remained more of a fundamental curiosity as opposed to a feature that can be applied in modern technologies. Due to their robustness against external perturbations, topological invariants in physical systems are increasingly being considered as a means to encode information. Hence, structured light with topological properties could potentially be used for such purposes. Here, we introduce the experimental realization of structures known as framed knots within optical polarization fields. We further develop a protocol in which the topological properties of framed knots are used in conjunction with prime factorization to encode information.

[1] Department of Physics, University of Ottawa, Ottawa, ON, Canada. [2] Research Laboratory of Electronics, Department of Electrical Engineering and Computer Science, Massachusetts Institute of Technology, Cambridge, MA, USA. [3] Center for Quantum Information Science and Technology, Faculty of Engineering Sciences, Ben-Gurion University of the Negev, Beersheba, Israel. [4] Faculty of Engineering, Institute of Nanotechnology and Advanced Materials, Bar Ilan University, Ramat Gan, Israel. ✉email: hlarocqu@mit.edu; ekarimi@uottawa.ca

Structured light—optical fields with shaped spatial and temporal features[1]—provides a viable platform for the realization of a variety of topological structures. The creation of such structures mostly draws from concepts related to singular optics[2,3], i.e., the study of discontinuities in optical wavefields. Such discontinuities, which can be present in features such as optical phase[4] or polarization[5], are known as optical singularities and can be employed to produce optical beams of varying complexity from those carrying a single singularity[6] to more exotic wavefields forming structures such as topological bands and knots. The latter include Möbius strips[7–10], multi-twist ribbons[11], knots within scalar optical fields[12–14], knotted topologies within bichromatic fields[15], and knots in polarization fields, which include both knotted electromagnetic field lines[16–20] and knotted polarization singularities[21]. These structured optical fields carrying topological features have found numerous applications in modern science. Most notably, optical beams with a single singularity, which include orbital angular momentum modes, have been extensively employed in high-dimensional quantum information[22] along with both classical[23] and quantum communications[24]. Knots, which are generally described as topologically classified arrangements of some closed filament[25], have also emerged as a promising framework to enable new forms of technologies. This promise is mostly attributed to knots having a braid representation, which is a cornerstone of topological quantum information[26–30]. However, in spite of their significant potential, knots created within optical fields[12–15,21] are mostly investigated in experiments within the framework of information theory in a similar way to simpler optical beams carrying a single singularity[31]. They are more than often treated as two-dimensional transverse optical modes, as opposed to a three-dimensional object defined by prospectively more useful topological invariants. This shortcoming arguably arises from a current lack of overlap between the fields of topological quantum information and singular optics—that is, optical topologies that can currently be realized in the laboratory cannot be readily used as a platform for existing topological information protocols and vice-versa.

In this article, we introduce and experimentally demonstrate the generation and observation of structures in optical polarization wavefields forming framed knots. We then use the latter as information carriers by means of a protocol devised to encode topological information through the conjoined usage of prime factorization and the knots' own topological invariants.

## Results

**Framed C-lines**. Knots ubiquitously describe how looped threads are arranged in space. For this reason, when analyzed within a physical framework, knots are typically found within fields defined by regions that unambiguously form curves in three-dimensional space. These knotted curves have been demonstrated in systems such as the vortices of fluids[32], the intensity nulls of scalar optical fields[12–14], and within the C-lines of optical polarization fields[21]. C-lines specifically consist of curves of pure circular polarization in monochromatic electromagnetic fields[33]. One of their most distinguishing features relates to the structure of the polarization field in their close proximity. Namely, they are enclosed by polarization ellipses with a major axis that rotates by integer multiples of $\pi$ along a closed contour surrounding the C-line. This trait is in display in Fig. 1a, b. For the case of paraxial optical beams, polarization is confined within the plane transverse to the beam's propagation, e.g., the $xy$ plane. As shown in Fig. 1a, this restriction constrains the plane over which this polarization axis rotation can be traced. Non-paraxial beams, however, can feature polarization vectors whose normal is not perpendicular to the beam's propagation. As displayed in Fig. 1b, this normal

vector in turn dictates the plane in which the axis of the ellipse completes a half rotation around the C-line. The presence of these rotations consists of the key structural feature considered while defining the framed knots reported in this work.

A framed knot in three-dimensional space is a knot, i.e., a looped curve, equipped with a vector field called a framing. The framing is nowhere tangent to the knot and is characterized by a number, the framing integer, which is the linking number of the image of the ribbon with the knot. In other words, it counts the number of times the vector field twists ($2\pi$ rotations) around the knot. Knotted ribbons generalize framed knots to an odd number of half-twists, e.g., knotted Möbius bands. Given the above definition, we define the framing of a closed C-line by the axis of the adjacent polarization ellipse whose axis is perpendicular to the C-line's tangent. This concept is illustrated in Fig. 1a, b, where we embolden the color of the polarization ellipse surrounding the C-line whose axis is perpendicular to its tangent, thereby defining its framing. In the rare case where all axes are perpendicular at a certain point of the C-line, the polarization vector defining the framing can be interpreted as the one enforcing its continuity with the least amount of twisting. This concept in turn defines the framing attributed to a knotted C-line. As illustrated in Fig. 1c, the latter may be constructed from a knotted field, $\mathbf{E}^k$, defined by a circularly polarized component, $E_-^k$, with knotted phase singularities, and a longitudinally polarized component, $E_z^k$, ensuring that $\mathbf{E}^k$ satisfies Maxwell's equations[34]. By superposing $\mathbf{E}^k$ with a plane wave with the opposite polarization helicity, $E_+^p$, knotted C-lines arising from the singular structure of $\mathbf{E}^k$ are created. As shown in Fig. 1d, e, increasing the amplitude of $E_+^p$ with respect to that of $E_z^k$ molds the resulting C-line into the knot formed by the phase singularities of $E_-^k$. Further discussions involving the dynamics of this process are provided in Supplementary Note 1. Note that $E_z^k$ is negligible for paraxial beams, which are the main experimental focus of this work. Hence, for such beams, the C-line aligns with the aforementioned knotted vortices regardless of the amplitude of $E_+^{p\,21}$.

**Braid representation**. In addition to their well-discernible three-dimensional structures, knots can also be represented by mathematical objects called braids. Geometrically, braids consist of intertwined arrangements of strands that do not turn back on ground that is already covered. Due to Alexander's theorem, every knot can be expressed as a closed braid. For instance, the trefoil knot shown in Fig. 2a can be expressed as the closure of the braid shown in Fig. 2b. The concept illustrated in these diagrams can be further extended to knots and braids formed in three-dimensional space. For example, the trefoil knot embedded in the torus shown in Fig. 2c can be obtained through a stereographic projection of the braid enclosed in the cylinder shown in Fig. 2d[14,35]. One way to perform this projection is to express this braid as the zeros of a complex field. This field is explicitly written as a function of the complex coordinates $(u, v)$, which relate to the spatial coordinates, $(x, y, h)$, in which the braid is embedded through $u = x + iy$ and $v = \exp(ih)$. This braided field can in turn be transformed into its corresponding knot with a stereographic projection defined by

$$u = \frac{\rho^2 + z^2 - 1 + 2iz}{\rho^2 + z^2 + 1}, \; v = \frac{2\rho e^{i\varphi}}{\rho^2 + z^2 + 1}, \qquad (1)$$

where $(\rho, \varphi, z)$ are the cylindrical coordinates of the three-dimensional space in which the knot is now embedded. In essence, this projection wraps the braid defined over $(x, y, h)$ into a knot in $(\rho, \varphi, z)$ by connecting its two ends, thereby effectively mapping the $h$ coordinate to $\varphi$[14]. Further discussions on how the

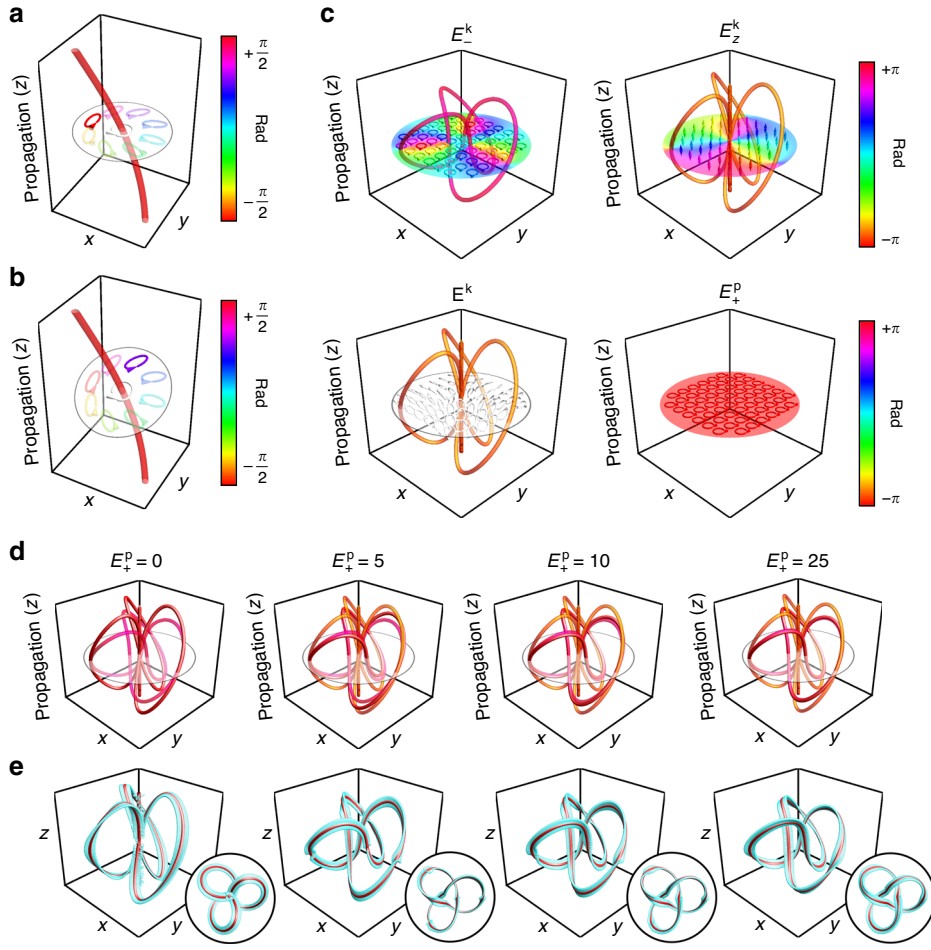

**Fig. 1 Construction of framed knotted C-lines.** Depiction of the polarization field in the proximity of a C-line when the normal of the polarization ellipse is **a** and is not **b** parallel to the beam's direction of propagation. Polarization ellipses with an axis perpendicular to the C-lines are displayed in bold colors and determine the orientation of the line's framing when it forms a closed loop. **c** Vector components of a framed optical knot, which include a circular component with knotted intensity nulls, $E_-^k$, accompanied by a longitudinal field, $E_z^k$, with nulls determined by the topology of $E_-^k$. These two components form a nonuniform polarization field $\mathbf{E}^k$ which can be shaped into a framed knotted C-line by means of a perturbing plane wave $E_+^p$. **d** Trajectory of the resulting knotted C-lines (red) overlaid onto the trajectories formed by the intensity nulls of $E_-^k$ (pink) and $E_z^k$ (orange) for various plane wave amplitudes. **e** Framed knot structures arising from the superpositions shown in **c** where the knotted C-line is shown in red and its frame is shown in cyan.

coordinates of each space map onto one another are provided in "Methods."

The above projection is heavily relied on when constructing knotted optical fields. In particular, a scalar optical field can be constructed by first matching its field along the $z = 0$ plane to that of the complex knot resulting from the projection of a braid as prescribed by Eq. (1). When this optical field is paraxial, then its formulation at subsequent $z$ planes can be obtained by means of paraxial propagation methods[14]. This method can then be further extended to describe paraxial-knotted C-lines[21] and full vectorial solutions to the optical wave equation[34]. For instance, the knotted field $E_-^k$ in Fig. 1c is fundamentally constructed based on the closure of a braid embedded within the zeros of a complex field[34].

Because of its wide usage in obtaining knots from braids, we have opted to use the projection defined in Eq. (1) to obtain structures with properties that can more easily be related to the braid representations of the optical-framed knots considered in this work. Namely, we consider the torus $\mathcal{T}_2$ obtained from the projection of the cylinder $\mathcal{C}$ enclosing the three-dimensional representation of the corresponding braid. Then, we scale the dimensions of our knots such that their structure fits within the proximity of $\mathcal{T}_2$. We later apply the coordinate transformation provided in "Methods" on those of a curve formed by a knotted

C-line. This transformation effectively cuts the knot along a given azimuthal angle and unwraps it, thereby mapping the $\varphi$ coordinate of the knot to the $h$ coordinate of the space where the braid is defined. During this process, the orientation of the knot's frame is assured to be locally preserved. To illustrate this procedure, we apply it on the framed optical trefoil knot shown in Fig. 2e. The resulting unwrapped structure is displayed in Fig. 2f. From this transformation, information such as the twisting angle in the knots' braid representations can be extracted. Here, the twisting angle consists of the azimuthal orientation of the ribbon in the frame where the normal is aligned to the unwrapped knot's tangent. For instance, the twisting angle in each strand of the unwrapped knot shown in Fig. 2f can be found in Fig. 2g.

**Prime encoding scheme.** Given the ability to extract the twisting angle of an optical-framed knot, we propose the following scheme exploiting these structures as information carriers. The use of this method relies on a pair of numbers $(\alpha, \beta)$ where $\alpha$ is a positive integer, and $\beta$ is a number both related to $\alpha$ and to the topological structure of the framed knot. The latter is given by

$$\beta = \prod_{\{k \mid d_k \neq -\infty\}} p_k^{\left(\alpha^{d_k - M}\right)}, \tag{2}$$

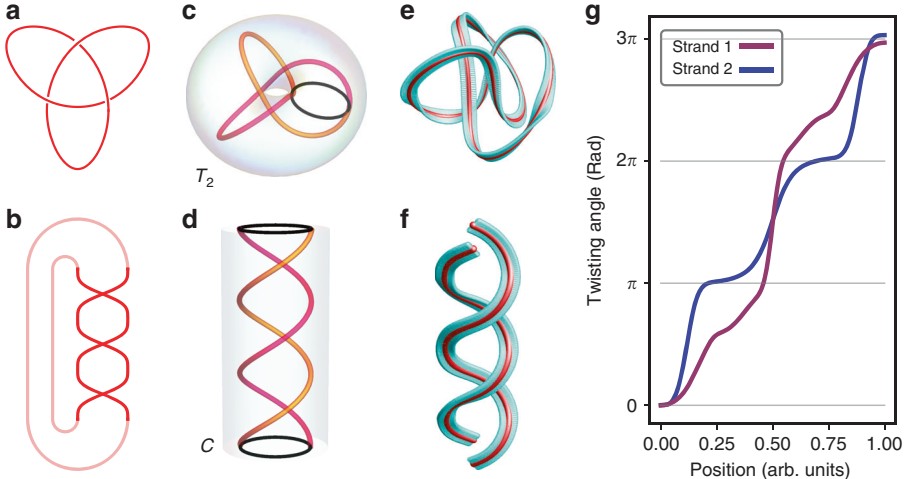

**Fig. 2 Braid representation of knots.** The diagram of a trefoil knot (**a**) along with its corresponding braid (**b**). Both ends of the braid diagram are connected to illustrate how its closure yields the knot in **a**. A trefoil knot (**c**) obtained from the stereographic projection of the braid in **d**. The projection effectively connects both ends of the braid, highlighted by a black outline, thereby transforming the two strands of the braid into a knot and the enclosing cylinder, $\mathcal{C}$, into a torus, $\mathcal{T}_2$. **e, f** An optical-framed knot (**e**) and its unwrapped form (**f**) obtained by applying a coordinate transformation on the curve formed by the knot while preserving the local orientation of the knot's frame. **g** Extracted twisting angle of the frame of the two strands in the structure shown in **f**.

where $k$ refers to a strand in a braid representation of the considered framed knot. $d_k$ is the number of half-twists along the $k$th strand exhibiting half-twists, i.e., $d_k = -\infty$ for untwisted strands. $p_k$ is a prime number assigned to the $k$th strand. Finally, $M = \Sigma_k d_k$ consists of the total number of half-twists in the knot's frame. Further discussions exploring how $\alpha$ and $\beta$ relate to braiding and twisting in framed braids are provided in Supplementary Notes 2 and 3. With these variables, we define the natural number

$$N_{\alpha,\beta}(M) \overset{\text{def}}{=} \beta^{(\alpha^M)} = \prod_{\{k|d_k \neq -\infty\}} p_k^{(\alpha^{d_k})}, \qquad (3)$$

whose prime factorization can be seen to be determined by the considered braid representation. Further details regarding this decomposition are provided in Supplementary Note 3.

The above representation of the framed knot and one of its braids may therefore be exploited for encoding and decoding topologically protected information as follows. Alice would like to send Bob a message which is here obtained as an output of a certain program running on some initial inputs, the set of numbers, $d_k$, $k = 1, 2, \ldots, n$. Running the program with this set is expected to yield Alice's message.

Alice conceives her program and its inputs as a framed braid. She identifies an operation with a sequence of crossings in the braid's planar diagram while the initial inputs are taken as the number of half-twists per strand. Alice has her program completely specified by the $n$-strand framed braid representation of a knotted ribbon $K_A$. To maintain some degree of privacy, she would like to send Bob $K_A$ rather than the original framed braid. As further discussed in Supplementary Note 3 and implied in Fig. 3, she takes note of the fact that $K_A$ may be complicated such as to conceal the original framed braid.

She then proceeds by performing the following steps. She first chooses a positive integer $\alpha$. She then determines the framed braid representation of $K_A$. Doing so involves allocating the number of half-twists in $K_A$ to different strands of the braid, i.e., setting $d_k$ such that $M_A = \Sigma_k d_k$. Following this step, she assigns prime numbers $p_k$ to strands exhibiting half-twists. Finally, she determines the number $\beta$ according to Eq. (2). Once this allocation is completed, Alice proceeds by sending Bob her knotted ribbon $K_A$ and the pair of numbers ($\alpha$, $\beta$) in real time.

Upon receiving these, Bob computes $N_{\alpha,\beta}(M_A)$ whose prime factorization unfolds $d_k$. To prevent the latter from being retrieved as an unordered set of integers, Alice and Bob rely on a previously adopted convention clarifying how the extracted $d_k$ is assigned to distinct strands of the encoded braid. Bob can now recover the framed braid that was originally considered by Alice. For illustrative purposes, we summarize this protocol in Fig. 3.

**Experimental generation**. Motivated by this encoding scheme, we proceed with its application to paraxial-knotted C-lines generated in the following experiments. Such structures can be created by means of the folded Sagnac interferometer used in ref. [21], which is shown in Fig. 4a for convenience. This apparatus separates a uniformly polarized light beam into two orthogonally polarized components, each of which modulated by a spatial light modulator (SLM). The latter displays holograms in which both the intensity and the phase of the target optical field is encrypted[36]. One component is modulated to produce a beam featuring knotted optical vortices[14], such as $E_-^k$ shown in Fig. 1c in the limit where non-paraxial effects are negligible. The other is modulated to form a large Gaussian beam that uniformly covers the entirety of the knotted component, thereby effectively taking the role of the plane wave $E_+^p$ in Fig. 1c. Upon exiting the interferometer, the two beams are coherently added, thereby converting the knotted phase vortices of $E_-^k$ into paraxial-knotted C-lines[21]. The knot and its frame can then be reconstructed with polarization tomography measurements[37] enabling one to obtain the field's polarization profile.

We use the above apparatus to produce both framed trefoil and cinquefoil knots. The holograms displayed on the SLM for this purpose are displayed in Fig. 4b along with the amplitude and phase of the fields that they are designed to generate. The latter are given in Eqs. (4) and (5) for the cases of the trefoil and cinquefoil knots, respectively,

$$\psi_{a,b,s}^{\text{Tref}}(\varrho, \varphi) = \left[ 1 - \varrho^2 - 4(a^2 - b^2)\varrho^3 - \varrho^4 + \varrho^6 \right.$$
$$\left. - 2(a - b)^2\varrho^3 e^{-3i\varphi} - 2(a + b)^2\varrho^3 e^{3i\varphi} \right] e^{-(\varrho/s)^2/2},$$
$$\qquad (4)$$

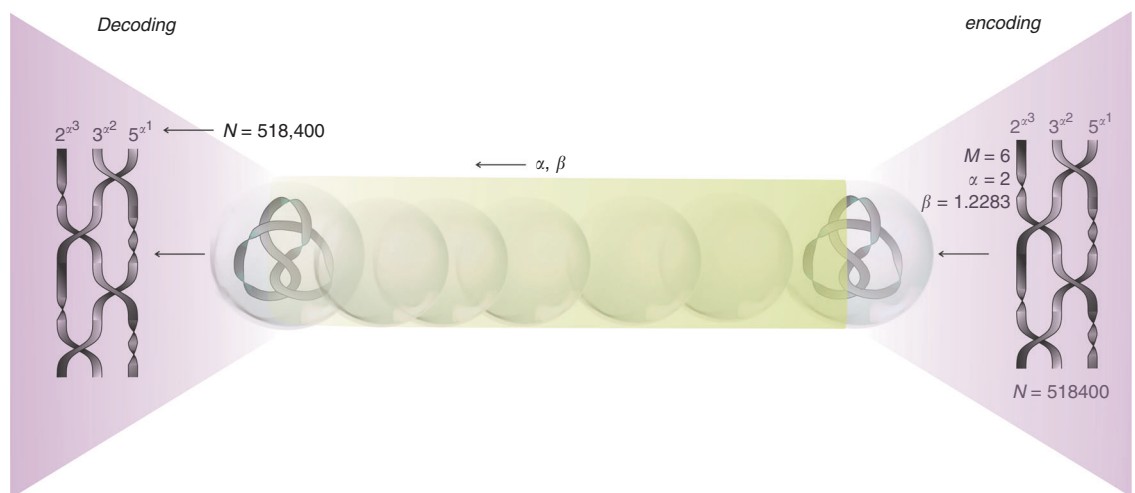

**Fig. 3 Prime encoding scheme of framed braids.** A framed braid on the right encodes a message—the output of a certain program specified by the planar diagram of the braid. In particular, the braid representation can be linked, as discussed below, to the prime factorization of a large integer $N$. An operation in such a program is identified with a sequence of crossings. Its inputs are taken as the number of half-twists per strand. To maintain privacy, the closure of the braid, i.e., the framed knot/knotted ribbon (in the case of even/odd number of half-twists, respectively), is transmitted instead of the braid itself. This allows the sender to complicate the message, if desirable, by adding an arbitrary number of Reidemeister-II and -III moves. The unique framed braid representation may be recovered on the receiver's end by transmitting two additional numbers, $\alpha$ and $\beta$, alongside with the knotted object. In this example, we chose for a simple elucidation $\alpha = 2$ and the first three primes $p_1 = 2$, $p_2 = 3$, $p_3 = 5$, one per strand in the braid (to showcase the scheme in the richer case of three strands, we preferred here the figure-eight knot, rather than the double-strand trefoil and cinquefoil knots). The corresponding numbers of half-twists in our example are $d_1 = 3$, $d_2 = 2$, and $d_3 = 1$, giving a total of $M = 6$ half-twists in the resulting framed knot. This is the topological invariant to be transmitted. The number $\beta$ is subsequently computed according to Eq. (2). Once received (on the left) the framed knot can be associated with the previously encoded integer $N$; the number of half-twists $M$ and the pair $\alpha$ and $\beta$ are substituted into Eq. (3) to yield $N$, which here equals 518, 400. The prime factorization of $N$ results from the actual number of half-twists per strand in the braid representation, $518, 400 = 2^{2^3} \cdot 3^{2^2} \cdot 5^{2^1}$.

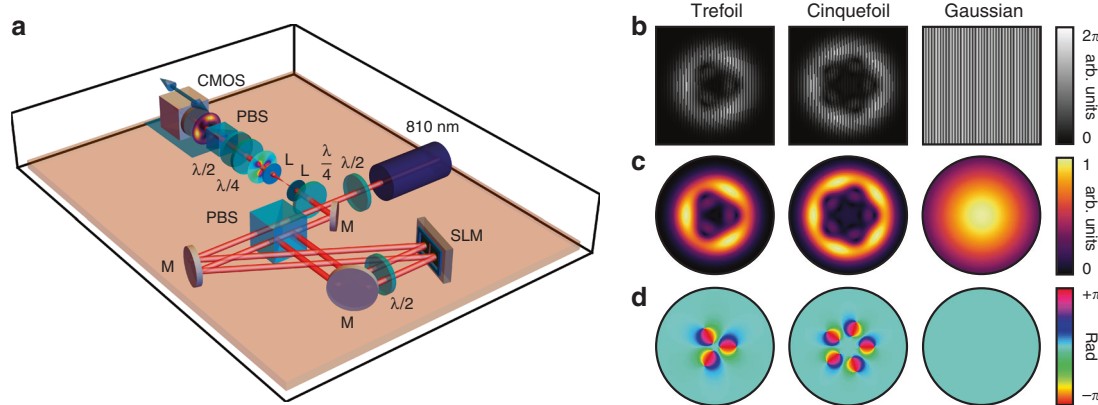

**Fig. 4 Experimental generation of optical-framed knots. a** Experimental apparatus used to generate optical-framed knots. An 810-nm laser produces light whose polarization is adjusted by a half-wave plate ($\lambda/2$) and later fed into a folded Sagnac interferometer. In the interferometer, the two polarized components are individually modulated by a spatial light modulator (SLM) and then coherently recombined to form an optical-framed knot. The latter is imaged using a 4f system, and then reconstructed by means of polarization tomography relying on a sequence of optical elements that include a quarter-wave plate ($\lambda/4$), a half-wave plate, a polarizing beam splitter (PBS), and a CMOS camera. Figure legend: mirror (M), L (lens). **b** Holograms used to generate framed knots, where knotted fields (trefoil and cinquefoil) are imprinted on the right-handed circular component of the optical field, and a Gaussian field is written on the left-handed component. **c** Amplitude and **d** phase of the fields generated by the corresponding holograms.

$$\psi_{a,b,s}^{\text{Cinq}}(\varrho, \varphi) = \big(1 + \varrho^2 - 2\varrho^4 - 16(a^2 - b^2)\varrho^5$$
$$- 2\varrho^6 + \varrho^8 + \varrho^{10} - 8(a-b)^2\varrho^5 e^{-5i\varphi} \qquad (5)$$
$$- 8(a+b)^2\varrho^5 e^{5i\varphi}\big]\ e^{-(\varrho/s)^2/2},$$

where $\varrho$ is a scaled and dimensionless version of the cylindrical radial coordinate, $\varphi$ is the azimuthal coordinate, and $a$, $b$, $s$ are parameters that determine the shape of the knot. For the trefoil knot, we considered parameters of $a = 1$, $b = 0.5$, and $s = 1.2$,

whereas for the cinquefoil knot, we used $a = 0.5$, $b = 0.24$, and $s = 0.65$. These fields are obtained based on stereographic projection methods explored in ref. [35] and are further discussed in Supplementary Note 4. As discussed in the latter, the selected parameters enable the creation of shorter knots. Furthermore, as emphasized in Supplementary Note 5, the frame of these knots is less disrupted by noise in the position of the C-lines arising from experimental imperfections. The framed knots of these fields expected from theory are shown in Fig. 5a, whereas the knots generated in our experiments can be found in Fig. 5b. Aside from minor perturbations that arise where the C-lines are born and

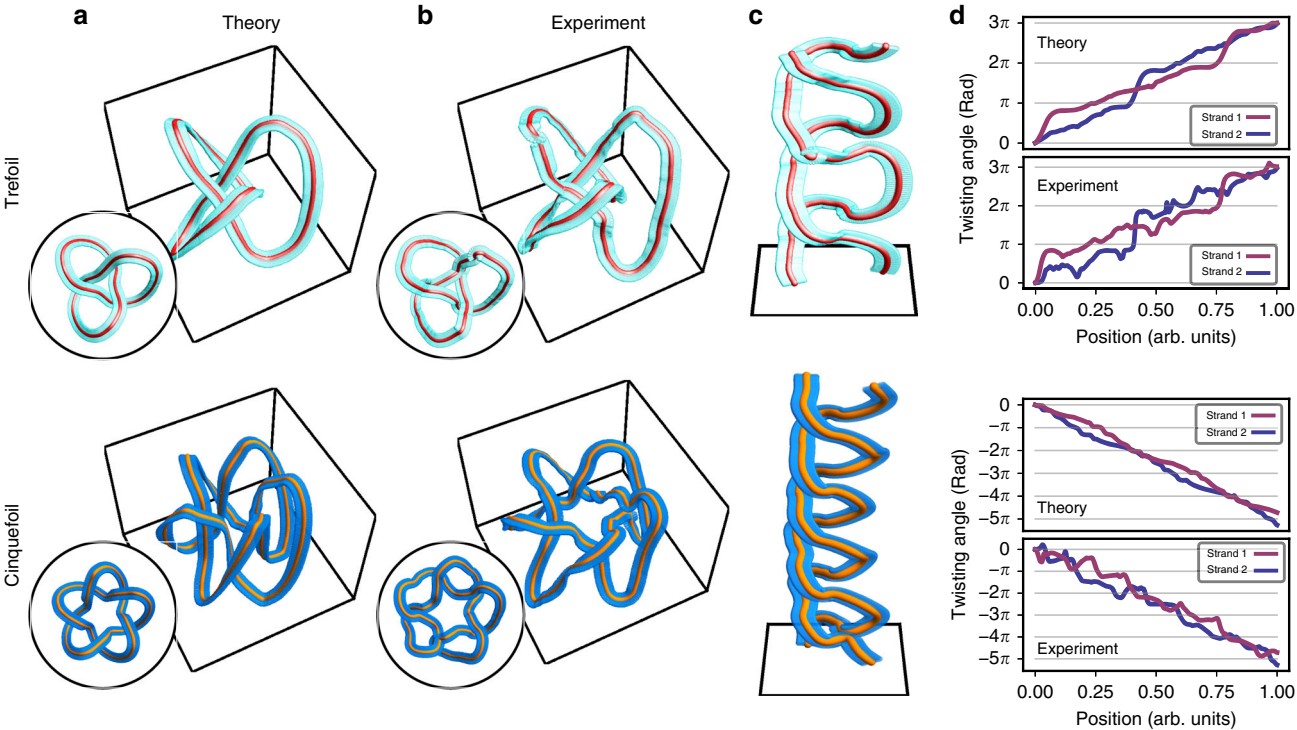

**Fig. 5 Trefoil and cinquefoil optical-framed knots. a** Optical-framed knots expected from theory attributed to the holograms displayed in Fig. 4. **b** Framed knots reconstructed from tomographic measurements of the beams generated by these holograms. **c** Unwrapped versions of the knots shown in **b**. **d** Twisting angle of the braid shown in **c** along with the values expected from theory.

annihilated at the knot's extremities, we observe that the knots' frames are in fairly good agreement with what is expected from theory. The unwrapped form of our experimental knots based on Eq. (1) is shown in Fig. 5c. We plot the corresponding twisting angle of these unwrapped knots along with the one expected from theory in Fig. 5d, where we observe once more that both strands in the structure are endowed with the same number of half-twists.

At this point, it is worth accentuating that the quantity of interest in Fig. 5d consists of the total twisting angle in the unwrapped knot. It might be tempting to treat the latter as one of the knot's braid representations. However, due to the knot's unwrapping, the number of half-twists in each strand may not exactly amount to an integer. Both ends of the braid are mapped from an azimuthal cross-section of the measured knot. Therefore, if the orientation of the frame at this cross-section is not the same for all parts of the knot, then the twisting angle of the strands in the unwrapped knot will not strictly amount to integer multiples of $\pi$. However, the sum of the twisting angles in each strand will amount to such a multiple given that the knot is a closed structure. This physical trait, in conjunction with the aforementioned $(\alpha, \beta)$ pair, in turn allows us to formulate the properties of the braid under consideration, which, for our purposes, consists of a purely algebraic entity. By taking this consideration into account and following the scheme outlined in Fig. 3, the knotted structures illustrated in Fig. 5 along with a given choice of $(\alpha, \beta)$ can be used to encode a braid representation of these knots.

## Discussion

The above generation and detection schemes can be extended to deal with non-paraxial optical knots. In practice, this extension would be achieved through the use of tight-focusing lenses and more sophisticated forms of polarization tomography[38]. As implied in Fig. 1, one could prospectively exercise further control over the C-line's frame with the presence of a stronger longitudinal polarization in the electric field. Furthermore,

non-paraxial methods would enable the generation of knots with a more manageable longitudinal extent. Indeed, a wealth of structures, including the trefoil and cinquefoil knots investigated here, are predicted to form over a distance comparable to the optical field's wavelength[34].

In practice, the act of sending information encoded within knotted C-lines by means of the scheme presented in this work could be achieved with an apparatus similar to the one presented in Fig. 4. The act of encoding information would be performed with the folded Sagnac interferometer enclosing the SLM and potentially other optics. Once the optical field is imbued with its knotted properties, it is then transmitted to a location where it can be decoded by means of an imaging system consisting of the two lenses shown in the setup. Finally, information is decoded from the field by means of the reconstruction techniques presented in this work, i.e., polarization tomography, to reconstruct the knot followed by a coordinate transformation to extract the corresponding braid. One could argue that the entirety of this information could be extracted from a single plane measurement of the field's properties, which would then enable the knot's reconstruction based on its theoretical propagation as prescribed by the optical wave equation. However, as alluded to in several parts of this work, sources of experimental imperfections such as aberrations, which have long been known to affect the topology of structured light beams[39], may potentially complicate such an approach. Namely, a full account of the aberrations and errors introduced by our interferometric setup would be needed to enable a full field reconstruction. This complication thereby encourages the use of the more direct reconstruction approach reported here.

To summarize, we introduced a construct for framed knots within optical polarization knots. This construct relies on the presence of knotted C-lines within the considered wavefield. Due to the singular nature of the axis of the polarization ellipse surrounding these lines, a frame can be assigned to the knot based on

its trajectory and the oscillation plane of the polarization ellipse along the C-line. Through the use of a coordinate transformation along with an encoding scheme relying on the braid representation of framed knots, we demonstrate how paraxial-knotted C-lines produced in experiments can be employed as information carriers. More specifically, we have shown that the braid representation of framed knots can be employed for encoding and decoding topologically invariant information such as the number of half-twists, as demonstrated above, being related to the prime factorization of large integers. It is expected that other topological invariants like the Alexander and Jones polynomials could consist of other encoded properties. The full potential of this scheme could be investigated by its application to more sophisticated knotted structures[35], such as experimentally generated figure-eight knots[21]. In addition, further engineering of the knot's frame through the use of non-paraxial structures[34,40] could be used to apply different numbers of half-twists within the knot. The generation of these more complex topologies may also be of interest in other applications relying on framed knots, such as quantum money[41,42]. On a more fundamental level, the methods outlined here may be of interest in quantifying and encoding the topological properties of more complex types of knots such as those that could be formed by the singularities in knotted tangles within random polarization fields[43] or speckle fields[5,44].

## Methods

**Coordinate mapping for the braid extraction of knots.** To unwrap the considered knots, we rely on the stereographic projection discussed in the main text. Based on the dependence of the complex variables $(u, v)$ on the spatial coordinates of the space in which the braid and knot are defined, which are respectively denoted as $(x, y, h)$ and $(\rho, \varphi, z)$, one can establish the following relation between both sets of coordinates

$$x \mapsto \frac{\rho^2 + z^2 - 1}{\rho^2 + z^2 + 1}, \qquad (6)$$

$$y \mapsto \frac{2z}{\rho^2 + z^2 + 1}, \qquad (7)$$

$$h \mapsto \varphi, \qquad (8)$$

where $(x, y, h)$ and $(\rho, \varphi, z)$ refer to Cartesian and cylindrical coordinate systems, respectively.

**Generation of optical-framed knots.** The generation of the knotted structures presented in this work relies on the method used in ref. [21]. As illustrated in Supplementary Fig. 1, an 810-nm laser is first coupled to free space where it later goes through a half-wave plate and a polarizing beam splitter (PBS) to modulate its intensity. It then goes through another half-wave plate in order to put the beam in an equal superposition of horizontal and vertical polarization components. The beam then goes through a folded Sagnac interferometer, which first separates each component with a PBS. In the interferometer, each polarization component is individually modulated by a SLM (Holoeye, Pluto Series) with the holograms shown in Fig. 4b. A half-wave plate is inserted within the interferometer to ensure that both components have the polarization that allows them to be modulated by the SLM. One of these components acquires a profile with knotted phase vortices[14] whereas the other acquires a large Gaussian profile. The two parts of the beam are then recombined at the exit of the interferometer where they are converted to circular polarizations by means of a half-wave plate (not shown in Fig. 4a) followed by a quarter-wave plate. In practice, this conversion can be achieved with only a quarter-plate as shown in Fig. 4a. However, we opted for the inclusion of the half-wave plate as it compensated for the effects of our dielectric mirrors on our beam's polarization, which needed to be introduced in our setup due to spatial constraints. As shown in ref. [21], converting the two polarizations to the circular polarization basis enables the conversion of knotted phase vortices into knotted C-lines.

The holograms provided in Eqs. (4) and (5) are expressed in terms of a dimensionless radial coordinate $\varrho = \rho/w_0$, where $\rho$ is the radial coordinate and $w_0$ is a scaling parameter. The $w_0$ values used to generate the knots shown in Fig. 4 were 0.35 mm for the trefoil knot and 0.42 mm for the cinquefoil knot.

**Measurement procedure.** We reconstruct the polarization field formed by the knotted C-lines by means of tomographic polarization measurements[37] at 40 planes spread out across the longitudinal extent of the knot. These measurements include six polarization projections along the horizontal, vertical, diagonal,

anti-diagonal, left- and right-handed circular polarizations. As shown in Supplementary Fig. 1 and depicted in Fig. 4a, the polarization projections are performed with a quarter-wave plate, followed by a half-wave plate, and then a PBS. The projections themselves correspond to the average of six frames recorded with a CMOS camera (Thorlabs DCC1645C).

Our choice of $w_0$ resulted in knots with an experimental longitudinal extent of 66 and 58 cm for the trefoil and cinquefoil knots, respectively. Measurement planes were separated by 1.5 cm while acquiring data for the trefoil knot whereas they were separated by 2 cm for the cinquefoil knot.

**Framed knot reconstruction.** Sources of image degradation in the polarization measurements, such as speckles of dust on the camera, are first removed by making the recorded measurements go through a non-aggressive low-pass filter. The processed images are then used to extract the beam's first, second, and third Stokes parameters, $s_1$, $s_2$, and $s_3$. The latter are employed to reconstruct the polarization profile of the knotted field at 40 planes transverse to the beam's propagation. As performed in ref. [21], the location of the C-lines along these planes is then determined by finding contour intersections in the phase of the field formed by $s_1 + is_2$. The transverse locations of the C-lines at each plane are then connected to provide the knot formed by the beam's polarization field. In order to smooth out numerical noise arising from the discretization of the CMOS images used to obtain the knot, a Gaussian filter with a width of 1 pixel is applied on the knot's transverse coordinates, i.e., $x$ and $y$. Spline interpolation is then used to estimate the knot's location outside of the considered transverse planes. The orientation of the frame of the knot is then extracted by taking the cross product between the gradient of the knot's coordinates and the normal vector of the C-line's oscillation plane. For the paraxial knots considered in our experiments, this orientation corresponds to the direction of propagation of the beam. More detailed discussions regarding the data processing involved in the reconstruction of the framed knot are provided in Supplementary Note 6.

## Data availability

The data that support the findings of this study are available from the corresponding author upon reasonable request.

## Code availability

The code producing the figures is available from the corresponding author upon reasonable request.

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

## Acknowledgements

This work was supported by Ontario's Early Research Award (ERA), Canada Research Chairs (CRC), and Canada First Research Excellence Fund (CFREF) Program. H.L. acknowledges the support of the Natural Sciences and Engineering Research Council of Canada (NSERC).

## Author contributions

H.L. designed the holograms used to generate the knots. H.L. and A.DE. performed the experiment. H.L. and A.DE. analyzed the data. H.L. and M.F.F.-G. performed the numerical simulations. A.C. and E.C. developed the prime encoding protocol. A.C., E.C., and E.K. supervised all aspects of the project. All authors discussed the results and contributed to the text of the manuscript.

## Competing interests

The authors declare no competing interests.
