## [Peer Review File · Nature Communications]

Reviewers' comments:

Reviewer #1 (Remarks to the Author):

The paper presents interesting ideas, which definitely meet the bar for publication, and the quality of the material is sufficient that I feel that the paper will eventually be suitable for Nature Communications. However, I feel that there are two deep issues that need to be addressed before this manuscript can be published: one is to do with presentation (in short, the presentation of prime-number encoding in the second half of the paper needs to be substantially improved) and the other is about the material itself (in short, I don't understand the role of polarization in the knot).

Major concern 1.

My first major concern is that the polarization properties of the beam appear to be superfluous to the construction of the knotted frame, and it seems to me that the same structures could be constructed using only the initial scalar knot. The authors' clearest statement of how the frame is constructed is in the SM (line 477): "the polarization ellipse surrounding the knot whose orientation is perpendicular to the knot's trajectory is identified and assigned to the knot's frame". As the authors note in line 72, the ellipse polarizations rotate by n around the C-line, so that all polarization directions are represented in the neighbourhood.

In particular, the knot frame used by the authors can be reconstructed using only the trajectory of the knot, as the chosen direction is given by the cross product between the direction of the trajectory and the unit normal vector to the plane that contains the polarizations (which is essentially the propagation direction vector for the paraxial approximation in use). In other words, at any point in the trajectory, the setup naturally sets up a plane (which contains the polarizations), and there is always one unique direction within this plane that is orthogonal to the knot's trajectory, regardless of what the polarization structures on that plane look like. As such, upon closer inspection, the polarization structure feels extraneous to me.

That said, the statement above, "there is always one unique direction..." is not strictly true, and this also impacts the construction as given in the paper: specifically, if the trajectory of the knot is along the normal to the plane of polarization (i.e. along the global direction of propagation), then no local frame direction can be chosen, because every direction in the plane (and the major axis of every polarization ellipse adjacent to the knot) is orthogonal to the trajectory of the knot. This is a major weakness of the scheme, and needs to be explicitly acknowledged in the manuscript.

Moreover, this concern extends to knot-trajectory directions which are close to this orthogonality condition, where the construction of the frame direction becomes an ill-conditioned numerical problem and essentially becomes unstable and subject to numerical noise: small changes to the knot's trajectory can have a large impact on the construction of the frame, up to and including the number of half-twists in that sector. (The authors explicitly recognize this, in lines 112-117, as the cause for the difference in twisting angle for their cinquefoil knots between theory and experiment in Figs. 2F and 2G.) This instability substantially weakens the topological-robustness argument for the use of these framed knots: yes, the topological invariant cannot be changed without breaking the frame, but the frame is fragile and easily broken.

While these concerns are not fatal by themselves, they require careful addressing by the authors before the manuscript can be published.

Major concern 2.

The other major problem with the current manuscript is that the second half of the paper, dealing with the use of the framed knots as a means to encode information and starting from line 118, is extremely unclear and hard to read; its presentation needs to be substantially overhauled to make it clear to the general audience that Nature Communications papers should address. At present, I find the bare essentials of the scheme to be completely opaque (and essentially incomprehensible) without referring to the Supplementary

Materials (which should categorically not be the case – the SM is a supplement, and the text should be readable on its own), and after going through the SM the scheme is still not completely clear to me.

I apologize for the harshness of this assessment, and I hate to be the bearer of bad news (particularly since the ideas in this paper are definitely interesting and should be published), but I suspect that the paper's general audience will have an even harder time with the text as currently written than I have. This paper is attempting a tough communication task, as there are virtually no readers fully conversant on both topological quantum computing and singular optics, and, as such, it needs to take its time with the concepts on both sides. I would encourage the authors to overhaul the presentation with clarity in mind, and taking the space as required (brevity is not a goal or a requirement here!), and then to test the manuscript on representative readers to ensure it is readable.

Minor points

3. The reference to "knots in polarization fields" to Ref. 12 in line 34 should really include a mention of the Hopf-Rañada fibration as well, ideally with a citation to the original Rañada papers (e.g. J Phys A 25, 1621 (1992) and Phys Lett A 202, 337 (1995)) as well as a suitable modern review.
4. The introduction of the concept of a framed knot, in lines 56-57, feels imprecise and hand-wavy. In brief, don't (just) tell me what the frame is "like", tell me what it *is*. A short and concise but precise definition of a framed knot is very much required here.
5. In particular, the definition in lines 56 and 57 seems to allow framed knots to have an odd number of half-twists, giving it a global Möbius-band structure (and therefore making the structure incompatible with the trivial fiber-bundle structure on line 359. If I understand the knot-theory literature correctly, this is indeed ruled out by the standard definition, but the vague definition offered by the current manuscript is not enough to discard this case.
6. The sentence 'the wavefields of the generated knots are obtained from a Milnor map of a braid onto the space in which the knot is formed', on lines 92-94, is a very sudden drop in readability due to the load of unexplained new terminology that is mostly not understood by this paper's likely audience, and needs to be explained at a slower pace.
7. The sentence 'the dark blue arrow provided in Figs. 2A,C preserves the local geometry of the space in which the knot and the braid are represented' is completely unclear to me.
8. Lines 120 and 384 mention 'subspaces' where presumably the authors meant tensor-product *factors*, which are not the same thing.
9. The authors present the claim, in lines 142 and 413, that the reconstruction of the braid representation is as hard as prime factorization, but I don't understand what the evidence is for this claim.
10. It would be good if the authors can explain the differences between the experimental setup pictured in Fig. 1C and the one used for Ref. (12), or confirm in the manuscript that it is the same apparatus.
11. In Figures 2D-G, I don't understand why the lines don't reach integer numbers of n at the right-hand side. Shouldn't they? Presumably there is at least *some* property of this graph that should add up to such a multiple, and it would be good if the manuscript explained what that is and how it is respected.
12. Figures 2D-G would be more readable if they were marked as trefoil/cinquefoil and experiment/theory on the figures themselves.
13. In Figures 3B and 3C, it is visually unclear to me whether the braid-crossing operations impose a twist on the frame or not (which is much clearer in Fig. S2). Here the fancy 3D graphics act rather against readability, I'm afraid.

14. The claims in line 348 (known algorithms to convert knots into braid representations scale polynomially) and 350 (there are multiple such braid representations) should be accompanied by suitable citations. Knot theory remains largely not understood by most physicists, and even elementary claims should contain links to material that can explain the relevant framework.

15. Figure S1 is labelled as showing the generators of FB_n , but it uses R instead of σ for the braid operation.

16. The Yang-Baxter relation in line 378 looks rather mystifying at first glance – but it becomes much clearer if one knows that R is a *two*-particle operator acting on two out of three of the relevant tensor factors. Similarly, that fact is essential to making the dimension counting in the displayed equation above make sense. It should be explicitly called out when R is introduced.

Overall, the paper is good, but my feeling is that it still needs a fair amount of work for it to be ready for journal publication.

Reviewer #2 (Remarks to the Author):

Topological structures in electromagnetic fields have been the subject of enormous amounts of work which have been mainly focused on purely fundamental questions such as the extent to which one can control light. While these explorations are essential to the advancement of the field, it is true, as the authors very well point out, that most of them have remained mere “curiosities”. Therefore, a work that presents a useful application of these structures would be of great interest. Now, this is what the authors aim to do in the present manuscript. They introduce the concept of framed knots along with a protocol to encode prime factorization of natural numbers and show that it can be realized in paraxial beams with a knotted C line. However, I feel that the framed knot structure is being forced upon the optical fields under study and does not appear as an intrinsic property. Because of this the encoded protocol feels disjointed from the optical implementation. Not to mention that optical knots are confined to a three-dimensional space and thus cannot be used to send information as opposed to what could be implied by the carefully chosen title (carrier is an ambiguous word). Whether information can be stored or encoded on them is a different matter, and this is what is being done in this work. Moreover, the protocol as it is currently explained in the text feels more like cryptography than anything else since there are number of parameters that need to be known a priori in order to retrieve the information. As it stands, I cannot recommend it for publication in Nature Communications. In what follows I provide further arguments to support my claims.

- The identification of the framed knots on optical fields is not clear. Framed knots are defined as knots for which a frame, forming “a vector field that is nowhere tangent to the knot’s trajectory”, exists. In their optical realization the knot is provided by the C-lines, just as the ones constructed in Ref. [12], and the frame is provided by the surrounding polarization structure since there is always an adjacent polarization that is orthogonal to the knot’s trajectory. This can mean two things: either the frame is given by the direction of the polarization that is orthogonal to the knot’s trajectory or the frame is given by the direction of the orthogonal polarization, similar to what is done to construct the Seifert surfaces. From what I can see in Fig. 1A it would seem to be the former. In which case, I would argue that this assignment is purely artificial since the frame would have been automatically fixed by the propagation direction which uniquely defines a transverse plane at each propagation distance. Now, this is irrespective of the surrounding polarization and could even be done in a scalar beam with the same knot structure. If this is not how the frame is assigned, then it should be clarified and explained how it is an intrinsic structure of the field and not artificially defined. Moreover, I would argue against the term “polarization direction” for referring to the direction defined for the mayor axis. I know that the term has been used before but to someone that has not encountered it before it can be a source of confusion.

- Framed knots are then used to encode an integer number through its prime factorization. However, there

are $n+1$ (n being the number of strands in the braid) parameters (the p_k 's and α) that need to be assigned in order to do so. These parameters then would need to be known in order for the encoded integer to be retrieved. Therefore, they seem to be playing the role of a key used to decode the information stored from the number of half twist in the braid representation of the knot. Moreover, it is mentioned that a framed knot can be represented by several framed braids therefore the braid representation that was used must be another known parameter in order to retrieve the encoded integer. Or the other way around, if the integer is known then the number of twists can be deduced by finding the prime factorization of the integer. In any case, it is not clear how this protocol would be useful in a practical setting and what sets it apart.

- The term information carrier is ambiguous in that it can be thought of an entity that can be used to send information. It is clear from the text that something along this line is thought by the authors since they say, "The complexity of this task therefore motivates the direct transmission of the coefficients d_k by using the knot as an information carrier" (lines 142-144). But I would argue that the knot is a structure lives in three-dimensional space created by the propagation of a beam, it is then fixed in space. A knot does not propagate. It would then be the transverse profile that would be sent with a way of forming the knot once received. But then all the information would be in the transverse profile. This comment also applies to the ellipsometry application mentioned in the Materials and Methods section. This application should be removed unless it is clarified what is meant by transmitting a knot through a material and exactly how the material properties can be deduced from the change in the encoded number. Evidently, all fields change when their propagating medium is changed but it is the relation between this change and the medium properties that make a technique useful for ellipsometry. This link has not been established and it is therefore a useless afterthought. Moreover, there are techniques that are far easier to implement while providing good and accurate results.

- The authors justify the topological invariants for their protocol due to their robustness against external perturbations. Yet, this is clearly not the case in the results presented. The cinquefoil knot is highly sensitive to variations of the transverse plane used for measuring the polarization structure. What I find surprising is that the authors do not even attempt to reconcile these two features.

These are my main reasons for not recommending its publication in Nature Communications. If the authors were to disagree with any of the points previously raised, then they should reconsider the presentation of their results. They should have a clear message that cannot be misconstrued in such a way.

Nonetheless, in what follow I include some other points that should be addressed should the editors decide that another round of review is warranted or if the authors decide to submit their work somewhere else.

- Are any of the previously studied ribbon polarization structures framed knots? Answering this might help clarify the definition of the framed knots in this work.
- The unwrapping of the framed knots could be better explained. Particularly the coordinate transformation mentioned in the text should be clearly identified in Fig. 2. Also, they should explain how they compute the twisting angle. This is important in relation to the discrepancies observed for the cinquefoil knot, since, intuitively one would expect the total twisting angle of the two braids to be a multiple of π since the framed knot forms a ribbon-like structure. Is this still the case for the experimental results (it is hard to tell from Fig. 2G) and if so then there were less twists in the framed knot.
- There is a missing figure number at line 364 and if the corresponding figure is Fig. S1 then the notation should be homogenized. Also, the number of braid generators is equal to the number of strands minus one, so $n-1$ not n as indicated in the equation after line 361 and it is the second to last relation that holds for $|i-j|>1$ no the last one.
- The caption of Fig. S2 does not correspond to the figures shown.

Reviewer #3 (Remarks to the Author):

The manuscript "Optical Framed Knots as Information Carriers" describes a novel experiment which realizes

framed knots in optical fields, and discusses their possible applications as means of storing information i.e. by encoding integers through their prime factorizations which can be mapped onto the number of twists in the representative braid.

On the whole, the experimental realization of optical knots, especially those with an additional framed structure, represents an important advance in physics, adding a new topological object to the list of experimentally realizable mathematical structure in real physical systems. However, I find that this manuscript is not currently suitable for Nature Communications unless it is substantially improved as follows.

1. While the experimental demonstration of knotted structures is undeniably a significant achieve, the main issue I have with the manuscript is that it contains very little details of the experiment. In fact, the experimental description comprises not much more than a sketch (Fig. 1c) and some highly processed data for comparison with theory (Fig 2), with almost no details of (i) what was actually measured, (ii) how exactly the experiment was performed beyond brief descriptions, (iii) photo of the experimental setup (iv) rigorous data analysis and/or statistical error analysis. In short, even the raw experimental results are missing, and the current presentation renders the results far from being reproducible. To be fair, some details on the hologram and reconstructed polarization profiles were shown in the Methods section. But even then, we will expect a manuscript at the level of Nature Communications to contain a far more detailed and rigorous presentation of the experiment.

2. The second part of the manuscript is concerned about the theory about representing an integer in terms of the framed braid of the knot, as mathematically described from lines 124-138. (i) Some of these ideas are not new, and the authors should give proper citations. (ii) It will be helpful for the general audience to make the math behind this section clearer. Actually, the logic is already quite clear to me, who has prior knowledge of knot theory. But to typical readers, it may be more intuitive to start with Eq 4 before presenting its relation of the framed braid (Eq 1). And above all, it will be instructive to have an accompanying basic example.

3. Related to the previous point, please check Fig S2. It states that there are three braids, where I only saw two. Also, some of the α 's don't seem to match the illustrations.

4. Two concerns about topological stability: (i) Since the same physical knot can correspond to many possible associated braids (and framed braids), to what extent is the prime factorization of a given integer unique for a given framed knot?
(ii) Even though different knots (framed or otherwise) are topologically distinct, are these different topological configurations physically stable and robust? How much energy does it typically take to transform from one to another? These are physical questions to be answered for any potential application.

5. What is the physical significance of the twisting angle plots in Fig 2? Although the different framed knots are topologically distinct, and their 3D representation for experiments vs theory seem to agree, the twisting angle plots don't seem to agree between experiment and theory?

Reviewer #1: The paper presents interesting ideas, which definitely meet the bar for publication, and the quality of the material is sufficient that I feel that the paper will eventually be suitable for Nature Communications. However, I feel that there are two deep issues that need to be addressed before this manuscript can be published: one is to do with presentation (in short, the presentation of prime-number encoding in the second half of the paper needs to be substantially improved) and the other is about the material itself (in short, I don't understand the role of polarization in the knot).

Reply: We would like to thank the reviewer for their very constructive comments on our manuscript. As outlined below, we have addressed the first issue by completely restructuring the prime encoding section and by also including a deeper introduction on the fundamental concepts behind it in the Supplementary. As for the second issue, we have extended our conceptual analysis of framed C-lines to three-dimensional non-paraxial fields. This extension ought to highlight the role of polarization in the knot.

Reviewer #1: Major concern 1.

My first major concern is that the polarization properties of the beam appear to be superfluous to the construction of the knotted frame, and it seems to me that the same structures could be constructed using only the initial scalar knot. The authors' clearest statement of how the frame is constructed is in the SM (line 477): "the polarization ellipse surrounding the knot whose orientation is perpendicular to the knot's trajectory is identified and assigned to the knot's frame". As the authors note in line 72, the ellipse polarizations rotate by π around the C-line, so that all polarization directions are represented in the neighbourhood.

In particular, the knot frame used by the authors can be reconstructed using only the trajectory of the knot, as the chosen direction is given by the cross product between the direction of the trajectory and the unit normal vector to the plane that contains the polarizations (which is essentially the propagation direction vector for the paraxial approximation in use). In other words, at any point in the trajectory, the setup naturally sets up a plane (which contains the polarizations), and there is always one unique direction within this plane that is orthogonal to the knot's trajectory, regardless of what the polarization structures on that plane look like. As such, upon closer inspection, the polarization structure feels extraneous to me.

Reply: The reviewer is absolutely correct in their assessment of the frame's construction. The fact that the circular polarization vector of the C-lines oscillates in the plane transverse to the beam's propagation also fixes the orientation of the frame within this plane. This therefore implies that the trajectories of the C-lines completely determine the frame of the knot and that a similar frame could be artificially constructed with a knotted scalar beam such as the one used in [1]. Besides providing a convenient means of tracking the knot as performed in [2], polarization itself does not play any crucial role in determining the knot's frame.

In order to address this limitation, we have modified the context in which the knot's optical structure is presented. Namely, instead of constraining ourselves to paraxial polarization fields, we now present the framed knot's construction based on non-paraxial optical beams. These beams can have a significant

longitudinal polarization component. The presence of this component implies that C-lines in these fields can oscillate outside of the plane transverse to the beam's propagation, thereby enabling a knotted C-line to have a frame that is not constricted within this plane. As a result, the frame is now determined by both the knot's trajectory and the orientation of the circular polarization vector of the C-lines. We schematically communicate this idea in Fig. 1a,b, which has been added in the revised version of the manuscript.

In order to conceptually explore this construction, we rely on a recently published construction of non-paraxial knot bundles [3]. These bundles illustrate how knotted circularly polarized fields give rise to topologically structured longitudinal fields. We use these fields to replace the knotted paraxial ones of the previous version of the manuscript. The other circular polarization is assumed to be a plane wave. An example of this pair of fields is depicted in Fig. 1b of the revised manuscript. In Fig. 1c,d, we illustrate how the relative amplitude of these two components affects the trajectory of the knot formed by the C-lines along with the orientation of its frame.

Though the fields used in Fig. 1 are based on polynomial beams, it clearly illustrates the role of polarization in constructing the knot's frame and lays down the groundwork leading towards our experimental results where the construction of the frame is trivial. Unfortunately, our research group is not currently equipped to generate and analyze complex knotted non-paraxial fields, thereby preventing any results that more strongly display the importance of the knot's polarization.

The entirety of the newly formatted **Framed C-lines** section of the Results should now clarify the above points.

Note that the construction of the knots in Fig. 1 are based on polynomial beams. Though these are not physical, the global dynamics of their singularities is expected to be maintained provided that they are physically truncated by a large enough aperture. These ideas are further explored in the discussions section of our work, where we provide holograms that can yield knotted C-lines by means of tight-focusing with a high NA lens:

The above generation and detection schemes can be extended to deal with non-paraxial optical knots. In practice, this extension would be achieved through the use of tight-focusing lenses and more sophisticated forms of polarization tomography [35]. As implied in Fig. 1, one could prospectively exercise further control over the C-line's frame with the presence of a stronger longitudinal polarization in the electric field. Furthermore, non-paraxial methods would enable the generation of knots with a more manageable longitudinal extent. Indeed, some structures are predicted to form over a distance comparable to the optical field's wavelength [32]. In Fig. 5, we display structures expected to form from the tight-focusing of a structured light beam. In the latter, we show both the trajectories formed by the phase vortices of each polarization component upon propagation, along with the trajectories of the C-lines formed by the polarization field resulting from the addition of these vectorial components. As respectively indicated in Figs. 5a,b, both trefoil and cinquefoil knotted topologies can be generated in the

C-lines of these tightly focused beams. Further details regarding the creation of these structures are provided in the Methods.

Reviewer #1: That said, the statement above, “there is always one unique direction...” is not strictly true, and this also impacts the construction as given in the paper: specifically, if the trajectory of the knot is along the normal to the plane of polarization (i.e. along the global direction of propagation), then no local frame direction can be chosen, because every direction in the plane (and the major axis of every polarization ellipse adjacent to the knot) is orthogonal to the trajectory of the knot. This is a major weakness of the scheme, and needs to be explicitly acknowledged in the manuscript.

Reply: Indeed, there might be some cases where the knot trajectory is aligned with the normal of the circular polarization vector, thereby allowing the frame to be defined by all possible orientations in these regions. In order to address this ambiguity, we recommend assigning a frame in such regions that experiences minimal variations while enforcing continuity between both ends. This thought is summarized in the following passage.

In the rare case where all axes are perpendicular at a certain point of the C-line, the polarization vector defining the framing can be interpreted as the one enforcing its continuity with the least amount of twisting.

Reviewer #1: Moreover, this concern extends to knot-trajectory directions which are close to this orthogonality condition, where the construction of the frame direction becomes an ill-conditioned numerical problem and essentially becomes unstable and subject to numerical noise: small changes to the knot’s trajectory can have a large impact on the construction of the frame, up to and including the number of half-twists in that sector. (The authors explicitly recognize this, in lines 112-117, as the cause for the difference in twisting angle for their cinquefoil knots between theory and experiment in Figs. 2F and 2G.) This instability substantially weakens the topological-robustness argument for the use of these framed knots: yes, the topological invariant cannot be changed without breaking the frame, but the frame is fragile and easily broken.

Reply: Such regions do indeed become problematic while dealing with experimental data. In the latest version of the manuscript, we have performed an entirely new set of experiments aiming to address this issue. Namely, we modified our holograms such that they would enable the generation of knots that are expected to be contracted along the beam’s direction of propagation. This contraction effectively increases the relative transverse motion of the C-lines, thereby reducing the degree to which the knot is close to the orthogonality relation mentioned by the reviewer and increasing the frame’s stability. Details about the formulation of the field of these new knots are now provided in the Supplementary (see Supplementary Notes 4 and 5). As shown in the newly formatted Fig. 5, we now observe very good agreement between experiment and theory.

Reviewer #1: While these concerns are not fatal by themselves, they require careful addressing by the authors before the manuscript can be published.

Reply: We hope that extending our conceptual framework to knotted non-paraxial beams and introducing compressed knots along the beam's propagation both clarifies the role of polarization in the construction of the knot's frame and presents a better case for the topological robustness of our structures.

Reviewer #1: The other major problem with the current manuscript is that the second half of the paper, dealing with the use of the framed knots as a means to encode information and starting from line 118, is extremely unclear and hard to read; its presentation needs to be substantially overhauled to make it clear to the general audience that Nature Communications papers should address. At present, I find the bare essentials of the scheme to be completely opaque (and essentially incomprehensible) without referring to the Supplementary Materials (which should categorically not be the case – the SM is a supplement, and the text should be readable on its own), and after going through the SM the scheme is still not completely clear to me.

Reply: The part dealing with the use of framed knots as a means to encode information has been significantly restructured in the latest version of the manuscript. Instead of a rushed discussion starting from the Hilbert space formulation of framed braids, we have adopted a more concise and self-contained description of our information encoding scheme. This description now follows after discussions on how the number of half-twists in a knot's framing is extracted in this work. Based on advice provided by the third reviewer, we then immediately introduce the α and β values on which the scheme relies and how they relate to a specific braid representation of the knot under consideration. This braid explicitly consists of the entity encoded within the framed knot. We then mention how α , β , and the number of half-twists determine a positive integer, whose prime factorization reveals the encoded braid. An illustration summarizing this scheme has also been included in what is now Fig. 3. This restructured component is fully contained within the **Prime encoding scheme** section of the revised manuscript.

We have opted to include further discussions related to the topic in the Supplementary, such as the ones pertaining to more advanced topics in knot theory and the Hilbert space representation of framed braids. Though these concepts are extremely useful in understanding the origin of the α and β values, we believe that they may confuse readers since they detract from the optics-focused tone of the manuscript.

Reviewer #1: I apologize for the harshness of this assessment, and I hate to be the bearer of bad news (particularly since the ideas in this paper are definitely interesting and should be published), but I suspect that the paper's general audience will have an even harder time with the text as currently written than I have. This paper is attempting a tough communication task, as there are virtually no readers fully conversant on both topological quantum computing and singular optics, and, as such, it needs to take its time with the concepts on both sides. I would encourage the authors to overhaul the presentation with clarity in mind, and taking the space as required (brevity is not a goal or a requirement here!), and then to test the manuscript on representative readers to ensure it is readable.

Reply: Once again, we would like to emphasize that we are extremely grateful for the reviewer's comments as it has sparked numerous discussions amongst the authors. The latter have allowed us to gain a better grasp on what we would like the manuscript to divulge while also making it accessible to both singular optics and topological computing audiences. We believe that the revised format of the current version of the manuscript now manages to perform this communication while also benefiting from a complete introduction of the ideas on which it is enabled by the "article" format of the manuscript (as opposed to the letter style in which it was previously written). Note that two authors have been added to the manuscript as they performed some work related to either new experiments or numerical simulations of non-paraxial knots. Both of these authors also have differing scientific backgrounds and were able to provide some valuable input on how to clearly disseminate the message of the revised manuscript.

Reviewer #1: Minor points

3. The reference to "knots in polarization fields" to Ref. 12 in line 34 should really include a mention of the Hopf-Rañada fibration as well, ideally with a citation to the original Rañada papers (e.g. J Phys A 25, 1621 (1992) and Phys Lett A 202, 337 (1995)) as well as a suitable modern review.

Reply: Our reference to Ref. 12 was originally targeted towards polarization fields exhibiting polarization singularities as opposed to ones referred by the reviewer involving knotted electromagnetic field lines. However, we agree that these works are relevant to the field and have now added the following list of references in the introduction:

J Phys A 25, 1621 (1992)
 Phys Lett A 202, 337 (1995)
 Nat Phys 4, 716 (2008)
 J Phys A 43, 385203 (2010)
 Phys Rev Lett 111, 150404 (2013)

The aforementioned passage in the introduction has now been replaced by:

and knots in polarization fields, which include both knotted electromagnetic field lines [14–18] and knotted polarization singularities [19].

Reviewer #1: 4. The introduction of the concept of a framed knot, in lines 56-57, feels imprecise and hand-wavy. In brief, don't (just) tell me what the frame is "like", tell me what it *is*. A short and concise but precise definition of a framed knot is very much required here.

Reply: The following definition of a framed knot, along with extensions on this structures known as knotted ribbons, is now included in the revised version of the manuscript:

A framed knot in three dimensional space is a knot, i.e., a looped curve, equipped with a vector field called a framing. The framing is nowhere tangent to the knot and is characterized by a

number, the framing integer, which is the linking number of the image of the ribbon with the knot. In other words, it counts the number of times the vector field twists (2π rotations) around the knot. Knotted ribbons generalize framed knots to an odd number of half-twists, e.g., knotted Möbius bands.

Furthermore, the following more technical definition of a framed knot has been included in the Supplementary:

In this work we chiefly care about framed knots and more broadly knotted ribbons (see Supplementary Figure 3). Mathematically speaking, they are embeddings of the solid torus in 3-sphere.

Definition 1 *A framed knot (K, V) in S^3 is a knot K , i.e., an embedding of S^1 , equipped with a vector field V called a framing.*

The framing is characterized by a number, the framing integer, which is the linking number of the image of the ribbon $I \times S^1$ with the knot. In other words, it counts the number of times the vector field twists (2π rotations) around the knot. Knotted ribbons generalize framed knots to an odd number of half-twists, e.g., knotted Möbius bands.

Reviewer #1: 5. In particular, the definition in lines 56 and 57 seems to allow framed knots to have an odd number of half-twists, giving it a global Möbius-band structure (and therefore making the structure incompatible with the trivial fiber-bundle structure on line 359. If I understand the knot-theory literature correctly, this is indeed ruled out by the standard definition, but the vague definition offered by the current manuscript is not enough to discard this case.

Reply: As mentioned in the previous comment, the definition of a framed knot should now exclude structures with an odd number of half-twists. The latter are now accounted for in the definition of knotted ribbons. After providing this definition, the terms “framed knots” and “framed ribbons” are used interchangeably for convenience.

Reviewer #1: 6. The sentence ‘the wavefields of the generated knots are obtained from a Milnor map of a braid onto the space in which the knot is formed’, on lines 92-94, is a very sudden drop in readability due to the load of unexplained new terminology that is mostly not understood by this paper’s likely audience, and needs to be explained at a slower pace.

Reply: This sentence references a technique [1] that has relatively been well explored over the last decade for the construction of knotted singularities in optical fields. However, given the broad audience that the manuscript aims to reach, we agree that this brief description of the knot’s structure may not be appropriate. For this reason, we have reiterated a detailed description of the knot’s construction in the latest version of the manuscript while also including a new figure (Fig. 2) that schematically depicts this construction. In essence, the knot is built from the closed braid to which it is attributed as prescribed by

Alexander's theorem. The braid itself is formed by the zeros of a complex field. Applying a stereographic projection onto this complex field effectively closes the braid, i.e. connects its two ends, thereby forming a complex field whose zeros form a knot. This procedure has been summarized in the following passage of the revised manuscript.

The concept illustrated in these diagrams can be further extended to knots and braids formed in three-dimensional space. For example, the trefoil knot embedded in the torus shown in Fig. 2c can be obtained through a stereographic projection of the braid enclosed in the cylinder shown in Fig. 2d [12, 36]. One way to perform this projection is to express this braid as the zeros of a complex field.

This field is explicitly written as a function of the complex coordinates (u,v) , which relate to the spatial coordinates, (x,y,h) , in which the braid is embedded through $u=x+iy$ and $v=\exp(ih)$. This braided field can in turn be transformed into its corresponding knot with a stereographic projection defined by

$$u = \frac{\rho^2 + z^2 - 1 + 2iz}{\rho^2 + z^2 + 1}, \quad v = \frac{2\rho e^{i\varphi}}{\rho^2 + z^2 + 1},$$

where (ρ,φ,z) are the cylindrical coordinates of the three-dimensional space in which the knot is now embedded. In essence, this projection wraps the braid defined over (x,y,h) into a knot in (ρ,φ,z) by connecting its two ends, thereby effectively mapping the h coordinate to φ [12]. Further discussions on how the coordinates of each space map onto one another are provided in the Methods.

The above projection is heavily relied on when constructing knotted optical fields. In particular, a scalar optical field can be constructed by first matching its field along the $z=0$ plane to that of the complex knot resulting from the projection of a braid as prescribed by Eq. (1). When this optical field is paraxial, then its formulation at subsequent z planes can be obtained by means of paraxial propagation methods [12]. This method can then be further extended to describe paraxial knotted C-lines [19] and full vectorial solutions to the optical wave equation [32]. For instance, the knotted field E^k in Fig. 1c, is fundamentally constructed based on the closure of a braid embedded within the zeros of a complex field [32].

Reviewer #1: 7. The sentence ‘the dark blue arrow provided in Figs. 2A,C preserves the local geometry of the space in which the knot and the braid are represented’ is completely unclear to me.

Reply: This statement refers to the fact that the stereographic projection that maps a complex braid to a complex knot effectively maps the aforementioned h coordinate of the space in which the braid is embedded to the azimuthal coordinate of the azimuthal φ coordinate of the space in which the knot is

embedded. This ‘dark blue arrow’ has been removed from figures altogether and should thereby not provide any further sources of unclarity. The aforementioned addition to the manuscript mentioning the construction of the knot based on the stereographic projection should now clarify the thought that sentence aimed to communicate.

Reviewer #1: 8. Lines 120 and 384 mention ‘subspaces’ where presumably the authors meant tensor-product *factors*, which are not the same thing.

Reply: The reviewer is correct. We have therefore removed any mentions of ‘subspaces’ in the main text and the Supplementary and replaced them with ‘spaces’. The fact that these spaces are tensor-product factors should also be made obvious by the notation that we use while describing them ($H \otimes H$).

Reviewer #1: 9. The authors present the claim, in lines 142 and 413, that the reconstruction of the braid representation is as hard as prime factorization, but I don’t understand what the evidence is for this claim.

Reply: What we intended to divulge in lines 120 and 384 was that recovering the braid conjointly encoded within the framed knot and the pair (α, β) involved the prime factorization of $\beta^\alpha M$, where M is the total number of half-twists in the knot. We now realize that our original statement might have come across as a general one regarding the complexity of extracting one of the braid representations of a framed knot. To address this, we now simply mention the task at hand as opposed to making any statement about the complexity of extracting the braid representation of a framed knot:

Finally, $M = \sum_k d_k$ consists of the total number of half twists in the knot’s frame. With these variables, we define the number

$$N_{\alpha,\beta}(M) \stackrel{\text{def}}{=} \beta^{(\alpha^M)} = \prod_{\{k|d_k \neq -\infty\}} p_k^{(\alpha^{d_k})},$$

whose prime factorization can be seen to be determined by the considered braid representation.

...

Upon receiving these, Bob computes $N_{\alpha,\beta}(M_A)$ whose prime factorization unfolds d_k .

Reviewer #1: 11. In Figures 2D-G, I don’t understand why the lines don’t reach integer numbers of π at the right-hand side. Shouldn’t they? Presumably there is at least *some* property of this graph that should add up to such a multiple, and it would be good if the manuscript explained what that is and how it is respected.

Reply: The lines do not strictly reach integer multiples of π given the method that we use to extract the twisting angle in the braid representation of the knot. In essence, it relies on using a coordinate transformation as defined by the stereographic projection mapping a braid into a knot. We use this coordinate transformation to partially reverse the projection used to obtain the knot in order to obtain the

unwrapped structures depicted in this work. The transformation maps the azimuthal coordinate of the knot to a longitudinal coordinate. Therefore, both ends of the braid are mapped from an azimuthal cross-section of the knot. If the frame at all points located in this cross-section are not the same, then the frame in each strand of the unwrapped knot will not amount to an inter multiple of π . However, the knot itself is a closed structure and therefore has a total twisting angle that amounts to such a multiple. Ultimately, this property is the only required trait for the encoding scheme reported in this work. This is clarified in the following addition to **Experimental generation** part of our **RESULTS** section

At this point, it is worth accentuating that the quantity of interest in Fig. 5d consists of the total twisting angle in the unwrapped knot. It might be tempting to treat the latter as one of the knot's braid representations. However, due to the knot's unwrapping, the number of half-twists in each strand may not exactly amount to an integer. Both ends of the braid are mapped from an azimuthal cross-section of the measured knot. Therefore, if the orientation of the frame at this cross-section is not the same for all parts of the knot, then the twisting angle of the strands in the unwrapped knot will not strictly amount to integer multiples of π . However, the sum of the twisting angles in each strand will amount to such a multiple given that the knot is a closed structure. This physical trait, in conjunction with the aforementioned (α, β) pair, in turn allows us to formulate the properties of the braid under consideration, which, for our purposes, consists of a purely algebraic entity.

Reviewer #1: 12. Figures 2D-G would be more readable if they were marked as trefoil/cinquefoil and experiment/theory on the figures themselves.

Reply: What used to be Fig. 2 has been replaced with Fig. 5. Everything pertaining to the trefoil knot is included in the top row of the figure, whereas material related to the cinquefoil knot is now at the bottom. *Theory* and *Experiment* labels have also been added to their respective reconstructed knots and twisting angle plots. This revised format ought to improve the readability of the figure.

Reviewer #1: 13. In Figures 3B and 3C, it is visually unclear to me whether the braid-crossing operations impose a twist on the frame or not (which is much clearer in Fig. S2). Here the fancy 3D graphics act rather against readability, I'm afraid.

Reply: What used to be Figure 3 has been removed from the latest version of the manuscript. However, newly added knot and braid diagrams that are explicitly made to depict the influence of half-twists and crossings have now been made simpler. Such diagrams can for instance be found in Fig. 3.

Reviewer #1: 14. The claims in line 348 (known algorithms to convert knots into braid representations scale polynomially) and 350 (there are multiple such braid representations) should be accompanied by suitable citations. Knot theory remains largely not understood by most physicists, and even elementary claims should contain links to material that can explain the relevant framework.

Reply: The sentence concerning the complexity of known algorithms for converting knots to braids no longer appears in the draft and so no reference was added. Equivalent braid representations are described in more detail in the Supplementary Information below Supplementary Figure 5.

Reviewer #1: 15. Figure S1 is labelled as showing the generators of FB_n , but it uses R instead of σ for the braid operation.

Reply: The notation attributed to the braiding (σ_j) and twisting (τ_j) operations has been made consistent in the revised version of the Supplementary. Note however, that we still use R while referring to the operator counterpart of σ .

Reviewer #1: 16. The Yang-Baxter relation in line 378 looks rather mystifying at first glance – but it becomes much clearer if one knows that R is a *two*-particle operator acting on two out of three of the relevant tensor factors. Similarly, that fact is essential to making the dimension counting in the displayed equation above make sense. It should be explicitly called out when R is introduced.

Reply: We now clarify that the R operator acts on $H^{\otimes 2}$:

where I is the identity operator, and R , a unitary operator on $H \otimes H$ that satisfies the Yang-Baxter equation,

Reviewer #1: Overall, the paper is good, but my feeling is that it still needs a fair amount of work for it to be ready for journal publication.

Reply: We hope that the above changes brought to the manuscript improved its readability and the way in which its ideas are presented.

Reviewer #2: Topological structures in electromagnetic fields have been the subject of enormous amounts of work which have been mainly focused on purely fundamental questions such as the extent to which one can control light. While these explorations are essential to the advancement of the field, it is true, as the authors very well point out, that most of them have remained mere “curiosities”. Therefore, a work that presents a useful application of these structures would be of great interest. Now, this is what the authors aim to do in the present manuscript. They introduce the concept of framed knots along with a protocol to encode prime factorization of natural numbers and show that it can be realized in paraxial beams with a knotted C line. However, I feel that the framed knot structure is being forced upon the optical fields under study and does not appear as an intrinsic property. Because of this the encoded protocol feels disjointed from the optical implementation.

Reply: As further elaborated below, we have extended our analysis to knotted singular polarization fields with a longitudinal field component. By doing so, the orientation of the polarization ellipse along the C-lines (which is fixed for paraxial knots), begins to vary and constrains the orientation of the knot’s frame to a plane that is not transverse to the beam’s propagation. This should now be explained in the newly structured **Framed C-line** part of the **RESULTS** section of the revised manuscript. We hope that

this addition will provide the sense that the frame is not forced on the optical field and that it is only trivialized for the case of the experimental paraxial fields presented in this work.

Reviewer #2: Not to mention that optical knots are confined to a three-dimensional space and thus cannot be used to send information as opposed to what could be implied by the carefully chosen title (carrier is an ambiguous word). Whether information can be stored or encoded on them is a different matter, and this is what is being done in this work.

Reply: Though knots are indeed embedded in three-dimensional space, their extent is confined within a certain length along the beam's direction of propagation. As mentioned by the reviewer, a sender can encode the knotted information within the optical beam by means of the SLM in what is now Fig 4a of the manuscript. We argue that the action of "sending" this information is accounted for by the imaging system consisting of the two lenses in our setup. The action of measuring the knot as performed in our experiments consists of the reception of this information. Of course, from a fundamental perspective, more than one photon is required to send this information which is not practical. We now clarify the aforementioned aspects of our scheme in the following passage of the Discussion section of the revised manuscript.

In practice, the act of sending information encoded within knotted C-lines by means of the scheme presented in this work could be achieved with an apparatus similar to the one presented in Fig. 4. The act of encoding information would be performed with the folded Sagnac interferometer enclosing the SLM and potentially other optics. Once the optical field is imbued with its knotted properties, it is then transmitted to a location where it can be decoded by means of an imaging system consisting of the two lenses shown in the setup. Finally, information is decoded from the field by means of the reconstruction techniques presented in this work, i.e. polarization tomography to reconstruct the knot followed by a coordinate transformation to extract the corresponding braid. One could argue that the entirety of this information could be extracted from a single plane measurement of the field's properties, which would then enable the knot's reconstruction based on its theoretical propagation as prescribed by the optical wave equation. However, as alluded to in several parts of this work, sources of experimental imperfections such as aberrations, which have long been known to affect the topology of structured light beams [38], may potentially complicate such an approach. Namely, a full account of the aberrations and errors introduced by our interferometric setup would be needed to enable a full field reconstruction. This complication thereby encourages the use of the more direct reconstruction approach reported here.

Reviewer #2: Moreover, the protocol as it is currently explained in the text feels more like cryptography than anything else since there are number of parameters that need to be known a priori in order to retrieve the information.

Reply: The protocol is indeed cryptographic given that it relies on extracting a braid from the knot generated from its closure (which are physically produced by means of the knots generated in this work),

the pair of numbers (α, β) , and the use of prime factorization. We believe that the frequent usage of the term “encoding” clarifies the intended use of the protocol.

Reviewer #2: As it stands, I cannot recommend it for publication in Nature Communications. In what follows I provide further arguments to support my claims.

Reply: We hope that the modifications that were brought in the revised version of the manuscript have clarified the way in which our work is communicated and have addressed the issues that were raised by the reviewer.

Reviewer #2: The identification of the framed knots on optical fields is not clear. Framed knots are defined as knots for which a frame, forming “a vector field that is nowhere tangent to the knot’s trajectory”, exists. In their optical realization the knot is provided by the C-lines, just as the ones constructed in Ref. [12], and the frame is provided by the surrounding polarization structure since there is always an adjacent polarization that is orthogonal to the knot’s trajectory. This can mean two things: either the frame is given by the direction of the polarization that is orthogonal to the knot’s trajectory or the frame is given by the direction of the orthogonal polarization, similar to what is done to construct the Seifert surfaces. From what I can see in Fig. 1A it would seem to be the former. In which case, I would argue that this assignment is purely artificial since the frame would have been automatically fixed by the propagation direction which uniquely defines a transverse plane at each propagation distance. Now, this is irrespective of the surrounding polarization and could even be done in a scalar beam with the same knot structure. If this is not how the frame is assigned, then it should be clarified and explained how it is an intrinsic structure of the field and not artificially defined.

Reply: Indeed, the frame is defined by the first of the cases mentioned by the reviewer. Namely, it is given by the direction of the polarization that is orthogonal to the knot’s trajectory. However, we argue that the assignment of the knot’s frame is not *artificial*, but is rather *trivial* given the paraxial nature of the beams that we generate. In the revised version of our manuscript, we emphasize this claim with the extension of our discussions on the knots’ frame to non-paraxial knots. Unlike paraxial knots, non-paraxial knots have a longitudinal electric field component, thereby implying that the circular polarization vectors of the C-lines do not necessarily oscillate in the plane transverse to the beam’s propagation. Therefore, to determine the knot’s frame, knowledge of the oscillation plane of the C-lines is required. These thoughts are summarized in what is now Fig. 1 of the revised manuscript.

Reviewer #2: Moreover, I would argue against the term “polarization direction” for referring to the direction for the mayor axis. I know that the term has been used before but to someone that has not encountered it before it can be a source of confusion.

Reply: We have now replaced all occurrences of “polarization direction” by “major axis” in our revised version of the manuscript.

Reviewer #2: Framed knots are then used to encode an integer number through its prime factorization. However, there are $n+1$ (n being the number of strands in the braid) parameters (the p_k 's and α) that need to be assigned in order to do so. These parameters then would need to be known in order for the encoded integer to be retrieved. Therefore, they seem to be playing the role of a key used to decode the information stored from the number of half twist in the braid representation of the knot. Moreover, it is mentioned that a framed knot can be represented by several framed braids therefore the braid representation that was used must be another known parameter in order to retrieve the encoded integer. Or the other way around, if the integer is known then the number of twists can be deduced by finding the prime factorization of the integer. In any case, it is not clear how this protocol would be useful in a practical setting and what sets it apart.

Reply: As mentioned in the reply to one of the comments made by the first reviewer, we have significantly modified the section of the manuscript presenting our protocol in order to clarify its intended usage. This modified section can now be found under the **Prime encoding scheme** header of the **RESULTS** section.

As clarified in this revised section, the goal of the protocol is not to encode an integer, nor the number of twists in the knot, but rather to encode a framed braid. This encoding requires three components. The first one is the framed knot. As demonstrated in this work, this is a physical entity whose topology conforms with that of the encoded braid. The next component is the α value, which is arbitrarily chosen. The last one consists of the β value, which depends on both the number of half-twists in the framed knots, M , and the chosen α value. Furthermore, it is constructed such that the prime factorization of $\beta^{\alpha M}$, which consists of a topological invariant of the structure under consideration, yields the encoded braid by means of the combined usage of the framed knot and the (α, β) pair.

The purpose of encoding a braid in the first place has been left as a more open-ended motivation. However, we now mention that this braid could be used as a program to perform a more specific task such as reading a transmitted message.

Reviewer #2: The term information carrier is ambiguous in that it can be thought of an entity that can be used to send information. It is clear from the text that something along this line is thought by the authors since they say, "The complexity of this task therefore motivates the direct transmission of the coefficients d_k by using the knot as an information carrier" (lines 142-144). But I would argue that the knot is a structure lives in three-dimensional space created by the propagation of a beam, it is then fixed in space. A knot does not propagate. It would then be the transverse profile that would be sent with a way of forming the knot once received. But then all the information would be in the transverse profile.

Reply: As mentioned in one of the earlier comments raised by the reviewer, the knot is confined within a length along the beam's direction of propagation and its "transmission" can be achieved by means of imaging techniques. This is indeed what is performed in the experimental realization of our knots displayed in Fig. 4 of the revised manuscript. The knot is encoded within an optical field by means of a spatial light modulator. The latter effectively modulates the beam such that it acquires the knotted beam's field profile at the waist plane ($z=0$). Therefore, right after hitting the SLM, there is a part of the knot that

is being formed. However, we emphasize that this is *not* where the knot is being measured. A 4f system consisting of two lenses is used to image the formation of this knot to a further point in our setup. This imaging system can be thought of as “sending” the knot to this further point where it can be measured.

Of course, the transverse profile of the knot determines how it will be formed upon the beam’s propagation. In practice, however, aberrations, misalignment, and other sources of imperfections can introduce subtle discrepancies in the transverse profile of the beam. These subtle discrepancies can in turn significantly affect the three-dimensional profile of a beam. Therefore, accounting for these discrepancies by means of the multi-plane measurement approach used in this work is more reliable than simply measuring the beam’s transverse properties at one plane. This claim ought to be reinforced by the minor differences between the structures expected from theory and the ones obtained from experiment in Fig. 5a,b of the revised version of the manuscript.

To summarize, the action of “sending the knot” consists of imaging it’s waist plane such that it can be reconstructed with sequential in-plane measurements along the knot’s longitudinal extent. The presence of experimental imperfections make this detection method much more reliable than a single measurement of the knot’s transverse profile at its waist. As addressed in a previous comment, this comment has been addressed in the discussions section of the work.

Reviewer #2: This comment also applies to the ellipsometry application mentioned in the Materials and Methods section. This application should be removed unless it is clarified what is meant by transmitting a knot through a material and exactly how the material properties can be deduced from the change in the encoded number. Evidently, all fields change when their propagating medium is changed but it is the relation between this change and the medium properties that make a technique useful for ellipsometry. This link has not been established and it is therefore a useless afterthought. Moreover, there are techniques that are far easier to implement while providing good and accurate results.

Reply: We have removed the ellipsometry discussions from our work.

Reviewer #2: The authors justify the topological invariants for their protocol due to their robustness against external perturbations. Yet, this is clearly not the case in the results presented. The cinquefoil knot is highly sensitive to variations of the transverse plane used for measuring the polarization structure. What I find surprising is that the authors do not even attempt to reconcile these two features.

Reply: Indeed the structures submitted in our original version were particularly susceptible to perturbations, which can be categorized into two parts. The first consists of perturbations that arise from regions of the knot where the C-lines are very close to being perpendicular to the plane in which the polarization ellipse oscillates. As mentioned in our reply to the comments of the first reviewer, deviations from theory due to numerical noise or very minor experimental imperfections can severely affect the frame in those regions. The second type of perturbation has a more fundamental nature. It relates to the fact that, unlike regular knots, the topology of framed knots is affected by the first Reidemeister move, which is illustrated below:

This move can effectively add a half twist to the knot's frame and can actually be brought to the knot by the introduction of spurious polarization singularities, introduced through imperfections, that interact with the ones of the knot.

In order to overcome the influence of these perturbations, we performed a new set of experiments in which we have modified the field of our framed knots. The use of these fields have effectively compressed the knot's longitudinal extent. As a result, the trajectories of the C-lines experience a transverse motion that is a lot more abrupt than what was used in our previous implementation. This abrupt motion considerably reduces the presence of C-lines that are perpendicular to the transverse plane, thereby causing the frame to be a lot more stable. Furthermore, one of the knot's circularly polarized components is endowed with a more anisotropic phase profile. This property, in conjunction with additional experimental precautions, seem to reduce the birth of polarization singularities that give rise to the first Reideister move, thereby further reducing perturbations on the knot's frame.

The aforementioned aspects pertaining to the above beam shaping aspects of the work have now been incorporated within Supplementary Note 5 and are briefly discussed in the following passage of the main text.

For the trefoil knot, we considered parameters of $a=1$, $b=0.5$, and $s=1.2$, whereas for the cinquefoil knot, we used $a=0.5$, $b=0.24$, and $s=0.65$. These fields are obtained based on stereographic projection methods explored in [36] and are further discussed in the Supplementary. As discussed in the latter, the selected parameters enable the creation of shorter knots. Furthermore, the frame of these knots is less disrupted by noise in the position of the C-lines arising from experimental imperfections.

Reviewer #2: These are my main reasons for not recommending its publication in Nature Communications. If the authors were to disagree with any of the points previously raised, then they should reconsider the presentation of their results. They should have a clear message that cannot be misconstrued in such a way.

Reply: We hope that the revised manuscript now addresses the issues raised by the reviewer. On a fundamental level, the assignment of the knot's frame is not *artificial*, but rather *trivial* given the nature of our experimental paraxial beams. As for the knot's transmission, the fact that the knot has a finite extent and can effectively be imaged from one point to the other allows it to be "sent" between two points. Aspects regarding our prime factorization method should also now be more concise. Finally, the new

experimental data featuring new field formulations should now address the issue of sources of perturbations on the knot's frame.

Reviewer #2: Nonetheless, in what follow I include some other points that should be addressed should the editors decide that another round of review is warranted or if the authors decide to submit their work somewhere else.

Are any of the previously studied ribbon polarization structures framed knots? Answering this might help clarify the definition of the framed knots in this work.

Reply: None of the previous implementations are framed knots. Furthermore, their nature is fundamentally different from the ones presented in this work. The ribbons are formed by the polarization major axis along a closed contour around the beam's axis. The ones in this work are based on the oscillation plane of a closed C-line.

Reviewer #2: The unwrapping of the framed knots could be better explained. Particularly the coordinate transformation mentioned in the text should be clearly identified in Fig. 2. Also, they should explain how they compute the twisting angle. This is important in relation to the discrepancies observed for the cinquefoil knot, since, intuitively one would expect the total twisting angle of the two braids to be a multiple of pi since the framed knot forms a ribbon-like structure. Is this still the case for the experimental results (it is hard to tell from Fig. 2G) and if so then there were less twists in the framed knot.

Reply: As mentioned in the reply to one of the comments of reviewer 1, this should now be clarified in the **Braid representation** part of the **RESULTS** section of the revised manuscript. This section discusses how the knot is constructed from a complex field with zeros forming the braid representation of the knot. By applying a stereographic projection to this field, one is able to obtain the knots generated in this work. Our method of obtaining our experimental knot's braid representation consists of reversing this projection and is discussed in the following addition:

The concept illustrated in these diagrams can be further extended to knots and braids formed in three-dimensional space. For example, the trefoil knot embedded in the torus shown in Fig. 2c can be obtained through a stereographic projection of the braid enclosed in the cylinder shown in Fig. 2d [12, 36]. One way to perform this projection is to express this braid as the zeros of a complex field. This field is explicitly written as a function of the complex coordinates (u,v), which relate to the spatial coordinates, (x,y,h), in which the braid is embedded through $u=x+iy$ and $v=\exp(ih)$. This braided field can in turn be transformed into its corresponding knot with a stereographic projection defined by

$$u = \frac{\rho^2 + z^2 - 1 + 2iz}{\rho^2 + z^2 + 1}, \quad v = \frac{2\rho e^{i\varphi}}{\rho^2 + z^2 + 1},$$

where (ρ, φ, z) are the cylindrical coordinates of the three-dimensional space in which the knot is now embedded. In essence, this projection wraps the braid defined over (x, y, h) into a knot in (ρ, φ, z) by connecting its two ends, thereby effectively mapping the h coordinate to φ [12]. Further discussions on how the coordinates of each space map onto one another are provided in the Methods.

...

Because of its wide usage in obtaining knots from braids, we have opted to use the projection defined in Eq. (1) to obtain structures with properties that can more easily be related to the braid representations of the optical framed knots considered in this work. Namely, we consider the torus T_2 obtained from the projection of the cylinder C enclosing the three dimensional representation of the corresponding braid. Then we scale the dimensions of our knots such that their structure fits with the proximity of T_2 . We then apply the coordinate transformation provided in the Methods on those of a curve formed by a knotted C -line. This transformation effectively cuts the knot along a given azimuthal angle and unwraps it, thereby mapping the φ coordinate of the knot to the h coordinate of the space where the braid is defined. During this process, the orientation of the knot's frame is assured to be locally preserved. To illustrate this procedure, we apply it on the framed optical trefoil knot shown in Fig. 2e. The resulting unwrapped structure is displayed in Fig. 2f. From this transformation, information such as the twisting angle in the knots' braid representations can be extracted. Here, the twisting angle consists of the azimuthal orientation of the ribbon in the frame where the normal is aligned to the unwrapped knot's tangent. For instance, the twisting angle in each strand of the unwrapped knot shown in Fig. 2f can be found in Fig. 2g.

The reason why each strand in the unwrapped knot does not exactly amount to a multiple of π and why this is not an issue is clarified in the following passage:

At this point, it is worth accentuating that the quantity of interest in Fig. 5d consists of the total twisting angle in the unwrapped knot. It might be tempting to treat the latter as one of the knot's braid representations. However, due to the knot's unwrapping, the number of half-twists in each strand may not exactly amount to an integer. Both ends of the braid are mapped from an azimuthal cross-section of the measured knot. Therefore, if the orientation of the frame at this cross-section is not the same for all parts of the knot, then the twisting angle of the strands in the unwrapped knot will not strictly amount to integer multiples of π . However, the sum of the twisting angles in each strand will amount to such a multiple given that the knot is a closed structure. This physical trait, in conjunction with the aforementioned (α, β) pair, in turn allows us to formulate the properties of the braid under consideration, which, for our purposes, consists of a purely algebraic entity.

Reviewer #2: There is a missing figure number at line 364 and if the corresponding figure is Fig. S1 then the notation should be homogenized. Also, the number of braid generators is equal to the number of strands minus one, so $n-1$ not n as indicated in the equation after line 361 and it is the second to last relation that holds for $|i-j|>1$ not the last one.

Reply: Both aspects mentioned by the referee have now been corrected. The missing figure number has now been added and the notation used in the Supplementary has now been homogenized. The indexing on the braiding operation now reaches $n-1$ instead of n .

Reviewer #2: The caption of Fig. S2 does not correspond to the figures shown.

Reply: Fig. S2 is no longer in the revised version of the Supplementary.

Reviewer #3: The manuscript "Optical Framed Knots as Information Carriers" describes a novel experiment which realizes framed knots in optical fields, and discusses their possible applications as means of storing information i.e. by encoding integers through their prime factorizations which can be mapped onto the number of twists in the representative braid.

On the whole, the experimental realization of optical knots, especially those with an additional framed structure, represents an important advance in physics, adding a new topological object to the list of experimentally realizable mathematical structure in real physical systems. However, I find that this manuscript is not currently suitable for Nature Communications unless it is substantially improved as follows.

Reply: We would like to thank the reviewer for sharing his positive outlook on our work. We hope that the modifications that were brought to the revised manuscript has increased its suitability for publication.

Reviewer #3: 1. While the experimental demonstration of knotted structures is undeniably a significant achieve, the main issue I have with the manuscript is that it contains very little details of the experiment. In fact, the experimental description comprises not much more than a sketch (Fig. 1c) and some highly processed data for comparison with theory (Fig 2), with almost no details of (i) what was actually measured, (ii) how exactly the experiment was performed beyond brief descriptions, (iii) photo of the experimental setup (iv) rigorous data analysis and/or statistical error analysis. In short, even the raw experimental results are missing, and the current presentation renders the results far from being reproducible. To be fair, some details on the hologram and reconstructed polarization profiles were shown in the Methods section. But even then, we will expect a manuscript at the level of Nature Communications to contain a far more detailed and rigorous presentation of the experiment.

Reply: Given that the experimental setup was virtually identical to the one used in our previous work, we originally had a loose description of it. We now clarify this fact in the revised version of the manuscript:

Such structures can be created by means of the folded Sagnac interferometer used in [19], which is shown in Fig. 4a for convenience. This apparatus separates a uniformly polarized light beam

into two orthogonally polarized components, each of which modulated by a Spatial Light Modulator (SLM). The latter displays holograms in which both the intensity and the phase of the target optical field is encrypted [37]. One component is modulated to produce a beam featuring knotted optical vortices [12], such as E_{-}^k shown in Fig. 1c in the limit where non-paraxial effects are negligible. The other is modulated to form a large Gaussian beam that uniformly covers the entirety of the knotted component. Upon exiting the interferometer, the two beams are coherently added, thereby converting the knotted phase vortices of E_{-}^k into paraxial knotted C-lines [19]. The knot and its frame can then be reconstructed with polarization tomography measurements [34] enabling one to obtain the field's polarization profile.

Furthermore, the Methods section now includes additional discussions pertaining to our setup, the measurements at hand, the way in which the experiment was performed, and how data was analyzed to obtain the main results of this work. These are summarized in the **Generation of optical framed knots** and **Measurement procedure** parts of the Methods. Our holograms are also now provided in Fig. 4b along with a depiction of their respective fields whose formulations are given in Eqs. (4,5).

The Supplementary also now includes a photo of the experimental setup. Though a statistical error analysis would be of interest here, it would involve a substantial amount of additional data to be taken. As mentioned in the addition brought to the Methods, the data required to reconstruct the knot's topology involves performing 6 automated polarization projections on the knotted beam (which involve rotating two waveplates) at 40 positions along the beam's direction of propagation for a total of 240 measurements. Given that our apparatus is not currently automated to record such a data set, taking measurements to perform a statistical analysis would require a substantial additional effort on our part. For this reason, we believe that such an analysis should be reserved for future work. However, in response to some of the comments of the first and second reviewer, we now discuss how the field formulation of our knots affects how susceptible it is to perturbations. This addition hopefully ought to provide the level of detail required for our work's publication and is discussed in depth in Supplementary Notes 4 and 5. Furthermore, in the revised version of the work, each polarization projection is obtained by averaging over 6 frames taken by the CCD camera used in this work, which should hopefully address some of the statistical elements raised by the reviewer. Supplementary Note 6 has also been added to provide a deeper explanation of how the raw data is processed to obtain the results shown in Fig. 5.

We would like to stress that this measurement approach is fairly standard in works relying on polarization tomography. More thorough error analyses are used a little more frequently in works where structured light is transmitted over a turbulent channel, such as underwater or within a metropolitan free-space link. The results of this analysis are mainly targeted at gauging the stability of the channel. Our experiments, however, are performed in a well-isolated laboratory. Performing a rigorous error analysis on our beams would in essence reveal the limits associated with the quality of our optics and the performance of our electronics, which are not of significant interest.

Reviewer #3: 2. The second part of the manuscript is concerned about the theory about representing an integer in terms of the framed braid of the knot, as mathematically described from lines 124-138. (i) Some of these ideas are not new, and the authors should give proper citations. (ii) It will be helpful for the general audience to make the math behind this section clearer. Actually, the logic is already quite clear to me, who has prior knowledge of knot theory. But to typical readers, it may be more intuitive to start with

Eq 4 before presenting its relation of the framed braid (Eq 1). And above all, it will be instructive to have an accompanying basic example.

Reply: (i) The proposed analogy of braids and prime factorization does not seem to appear elsewhere in the relevant literature. However, somewhat related concepts may be found in Ref [7] now cited within the Supplementary Information, and in

Sundance Bilson-Thompson, Jonathan Hackett, and Louis H Kauffman. Particle topology, braids, and braided belts. Journal of Mathematical Physics, 50(11):113505, 2009.

Sundance Bilson-Thompson, Jonathan Hackett, Lou Kauffman, and Lee Smolin. Particle identifications from symmetries of braided ribbon network invariants. arXiv preprint arXiv:0804.0037, 2008

Gresnigt, N. (2018). Knotted boundaries and braid only form of braided belts. arXiv preprint arXiv:1808.03910.

where braided belts and ribbons are related to elementary particles. These references have been added to the Supplementary.

(ii) As mentioned in the reply to comments made both by reviewers 1 and 2, the part of the manuscript presenting the theory behind the encoding protocol has been significantly reworked and now falls entirely within the **Prime encoding scheme** part of the **RESULTS** section.

As suggested by the reviewer, we have started with Eq. (4) (which is now Eq. (2) in the revised version of the manuscript). We do so shortly after defining the framed knot and the α value as the first two required quantities for encoding a framed braid. Eq. (2) then presents the β as the last quantity required to do so. The section now proceeds by explaining how the combined use of these three entities can be used to compute the topological invariant $N_{\alpha,\beta}(M) = \beta^\alpha M$ whose prime factorization yields the encoded framed braid.

We have provided a basic example of this encoding scheme in what is now Fig. 3. In order to depict how the interaction of multiple strands in the braid can result in rich features yielding a large $N_{\alpha,\beta}(M)$, we have opted to use the braid representation of a figure-8 knot for this illustrative example.

Reviewer #3: 3. Related to the previous point, please check Fig S2. It states that there are three braids, where I only saw two. Also, some of the α 's don't seem to match the illustrations.

Reply: The Supplementary has been restructured such that Fig S2 is not included anymore. We initially accidentally mislabeled 'strands' as braids. We made sure that this misuse of terminology was absent in the revised version of the manuscript.

Reviewer #3: 4. Two concerns about topological stability: (i) Since the same physical knot can correspond to many possible associated braids (and framed braids), to what extent is the prime factorization of a given integer unique for a given framed knot?

(ii) Even though different knots (framed or otherwise) are topologically distinct, are these different topological configurations physically stable and robust? How much energy does it typically take to transform from one to another? These are physical questions to be answered for any potential application.

Reply: The degree to which the prime factorization of a given integer is unique for a given framed knot should now be clarified in the following passage that was added in the Supplementary

As previously noted, the braid representation of a knot is not unique. Braid representations of the same knot are related by two kinds of Markov moves known as stabilization and conjugation. The same applies to knotted ribbons as long as the number of half-twists remains unchanged by these moves. As shown in Supplementary Figure 6, this invariance is carried over to the prime factorization captured by the framed braid. The left identity is stabilization -- the act of adding untwisted strands, possibly interacting through Reidemeister-I moves, -- which here amounts to padding the prime factorization with 1's, i.e., no half-twists are introduced by the newly added strands (primes). Note that there must be as much Reidemeister-I moves as their inverses for otherwise the number of half-twists would change. Similarly, conjugation of a braid A , i.e. the newly formed braid, BAB^{-1} on the right of Supplementary Figure 6 below, where $BB^{-1} = B^{-1}B = I$ is the framed unbraided, has the same number of half-twists as A , hence also the same prime factorization.

As for the topological robustness of the generated structures, their vulnerability to external perturbations ultimately depend on the physical system in which they are produced. In our case, the knots are generated within the polarization singularities of a structured light beam, thereby making them susceptible to perturbations such as aberrations in the optics used in our experiments. As mentioned in some earlier comments, we have addressed this issue by adopting holograms that can generate optical beams whose singular dynamics are less susceptible to perturbations. These thoughts are summarized in the following passage:

For the trefoil knot, we considered parameters of $a=1$, $b=0.5$, and $s=1.2$, whereas for the cinquefoil knot, we used $a=0.5$, $b=0.24$, and $s=0.65$. These fields are obtained based on stereographic projection methods explored in [36] and are further discussed in the Supplementary Information. As discussed in the latter, the selected parameters enable the creation of shorter knots. Furthermore, the frame of these knots is less disrupted by noise in the position of the C-lines arising from experimental imperfections.

The additional discussions regarding the robustness of the generated structures are provided in Supplementary Note 5 of the Supplementary.

Aberrations have long been known to disrupt topological entities realized within structured light beams. This thought is reiterated in the following passage which can now be found in the Discussion section:

However, as alluded to in several parts of this work, sources of experimental imperfections such as aberrations, which have long been known to affect the topology of structured light beams [38], may potentially complicate such an approach. Namely, a full account of the aberrations and errors introduced by our interferometric setup would be needed to enable a full field reconstruction. This complication thereby encourages the use of the more direct reconstruction approach reported here.

However, in order to confine the scope of this work to something manageable for a general audience, we would like to reserve a more quantitative analysis of the influence of aberrations on the topological structures generated here for future work.

Reviewer #3: 5. What is the physical significance of the twisting angle plots in Fig 2? Although the different framed knots are topologically distinct, and their 3D representation for experiments vs theory seem to agree, the twisting angle plots don't seem to agree between experiment and theory?

Reply: As discussed in a comment addressed to the second reviewer, the source of disagreement between theory and experiment in regards to the twisting angle arises from sources of perturbations affecting the knot's frame. On one hand, C-lines that are perpendicular to the oscillation plane of the circular polarization vector are very susceptible to both experimental and numerical noise, thereby adding half-twists to the knot. On the other hand, spurious singularities interacting with those forming the intended knot can effectively perform the first Reidemeister move on the knot's topology. This adds an additional half twist to the knot. Both of these sources of perturbations are the cause of the discrepancies between the twisting angles expected from theory and those measured from experiment.

As mentioned earlier, we addressed this issue by reformulating our knotted fields such that they would be more resilient to these perturbations. Discussions regarding this formulation and its robustness are now included in Supplementary Notes 4 and 5.

References:

- [1] Dennis, M.R., King, R.P., Jack, B., O'Holleran, K. and Padgett, M.J., 2010. Isolated optical vortex knots. *Nature Physics*, 6(2), pp.118-121.
- [2] Larocque, H., Sugic, D., Mortimer, D., Taylor, A.J., Fickler, R., Boyd, R.W., Dennis, M.R. and Karimi, E., 2018. Reconstructing the topology of optical polarization knots. *Nature Physics*, 14(11), pp.1079-1082.
- [3] Sugic, D. and Dennis, M.R., 2018. Singular knot bundle in light. *JOSA A*, 35(12), pp.1987-1999.
- [4] Dennis, M.R., Götte, J.B., King, R.P., Morgan, M.A. and Alonso, M.A., 2011. Paraxial and nonparaxial polynomial beams and the analytic approach to propagation. *Optics letters*, 36(22), pp.4452-4454.

REVIEWERS' COMMENTS:

Reviewer #2 (Remarks to the Author):

After reading the title and abstract of the manuscript, and despite the critical tone of my previous review, I was excited to understand how these three-dimensional structures were used in a meaningful way. But I was let down by an opaque presentation that made it difficult to grasp the authors' message: neither the definition of the framed knot in terms of the polarization nor the prime factorization encoding were properly explained. Therefore, I am glad to see that the authors took the time to properly address all of the reviewers' concerns and criticism leading to a significant improvement in readability. I now think this work will be suitable for publication after some minor comments are addressed.

The definition of the frame using the paraxial and nonparaxial cases shown in Fig.1 makes it clearer and shows that it is not forced onto the fields. Therefore, I do not think the paragraph introducing the framed knots in nonparaxial fields (lines 72-90) is necessary in the main text since it is not essential to understand the main results. I suggest moving it to supplementary materials as well as making the following changes:

- The notation of the fields should be homogenized with the supplementary note 3: e.g. the superindex k (which I assume stands for knot) is not used in the latter.
- Does the superindex p stand for planewave? If so, why not use k since it is used to construct the knot?
- While the discussion about the role of E_{p+} in the knot structure is interesting, its relevance to the current work is not clear. Does the field E_{p+} affect the stability and/or size of the knot? And how do you choose its value? Is it related to the Gaussian beam used in the experimental implementation?
- It is hard to see what is going on in Fig. 1c-e; they are too small. The scale used for the axes should also be indicated.
- For the knotted lines in Fig. 1d it might be worth adding the case $E_{p+}=0$.
- Given that most of the nonparaxial treatment is done with polynomial beams, the results shown in Fig. 6 seem like an afterthought. The generation of optical knots with focused fields using Richards-Wolf theory has already been considered in Ref. [32]. Therefore, unless the effect of focusing on the frame of the knot is discussed, there is no need for Fig. 6.

Concerning the prime encoding scheme, I have some minor comments:

- The parameter α should be a positive integer and not a positive real number, right? If not, N would not be a natural number.
- Once Bob receives the pair (α, β) and the framed knot, he then proceeds to measure M , the total number of half twists in the framed knot. With these three parameters and their relation to the dk parameters he can retrieve them. However, they are retrieved as an unordered set of integers which can be used to recover an equivalent framed braid representation. Unless I am wrong, this should be stated explicitly in the main text. Right now it is only alluded at the very end of Supplementary note 2.

Other minor comments:

- Freund was the first to show the existence of Mobius bands in optical fields, so his work [e.g. Opt. Commun. 283, 1-15 (2010) and Opt. Lett. 35, 148-150 (2010)] should be cited alongside Refs. [7,8].
- In Sup. Fig. 7 it is hard to tell that the knots become shorter with b .

Reviewer #3 (Remarks to the Author):

In the revised manuscript, the authors have substantially improved the presentation of the manuscript, particularly on the prime encoding scheme with framed knots and their experiment.

As such, they have in the process addressed all my concerns, many of which also overlaps with Reviewer #1's.

With the more substantial experiment description of among the first practical application of optical knots, I

now consider their work ready for publication in Nature Communications. As far as I know, this is the first experimental work that illustrates a potential application of a type of topology different and much more sophisticated than the usual topological invariants of bandstructures i.e. Chern numbers and Z_2 indices.

Reviewer #2: After reading the title and abstract of the manuscript, and despite the critical tone of my previous review, I was excited to understand how these three-dimensional structures were used in a meaningful way. But I was let down by an opaque presentation that made it difficult to grasp the authors' message: neither the definition of the framed knot in terms of the polarization nor the prime factorization encoding were properly explained. Therefore, I am glad to see that the authors took the time to properly address all of the reviewers' concerns and criticism leading to a significant improvement in readability. I now think this work will be suitable for publication after some minor comments are addressed.

Reply: We are glad to hear that the revised version of our manuscript clarified its content and hope that our edits addressing the reviewer's minor comments in this round further elucidate the goal of our work.

Reviewer #2: The definition of the frame using the paraxial and nonparaxial cases shown in Fig.1 makes it clearer and shows that it is not forced onto the fields. Therefore, I do not think the paragraph introducing the framed knots in nonparaxial fields (lines 72-90) is necessary in the main text since it is not essential to understand the main results. I suggest moving it to supplementary materials as well as making the following changes:

Reply: We have trimmed what originally consisted of lines 72-90 down to a much shorter passage that minimally explains the content of Fig. 1c-e. This new passage is provided below:

This concept in turn defines the framing attributed to a knotted C-line. As illustrated in Fig. 1c, the latter may be constructed from a knotted field, \mathbf{E}^k , defined by a circularly polarized component, E_z^k , with knotted phase singularities, and a longitudinally polarized component, E_z^k , ensuring that \mathbf{E}^k satisfies Maxwell's equations [34]. By superposing \mathbf{E}^k with a plane wave with the opposite polarization helicity, E_+^p , knotted C-lines arising from the singular structure of \mathbf{E}^k are created. As shown in Fig. 1d,e, increasing the amplitude of E_+^p with respect to that of E_z^k molds the resulting C-line into the knot formed by the phase singularities of E_z^k . Further discussions involving the dynamics of this process are provided in Supplementary Note 3. Note that E_z^k is negligible for paraxial beams, which are the main experimental focus of this work. Hence, for such beams, the C-line aligns with the aforementioned knotted vortices regardless of the amplitude of E_+^p [21].

The old passage has also been moved to Supplementary Note 3. Given that some of its content was already provided in the Supplementary, the passage has been modified as follows

At this point, a few remarks can be made regarding the interplay between the knotted field \mathbf{E}^k and the plane wave E_+^p . On its own, \mathbf{E}^k has C-lines located by default along the nulls of the E_z^k field. The coherent addition of E_+^p deforms the C-line into a knot within the proximity of the one formed by E_z^k . When the amplitude of this additional field is increased, the knotted C-line gradually adopts the shape of the E_z^k knot and its frame experiences less influence from E_z^k . The formation of the C-lines in \mathbf{E}^k and how they are deformed by the addition of E_+^p are depicted in Figs. 1c,d of the main text, respectively. The normal of the circular polarization vector then gradually aligns itself to the beam's direction of propagation, z , thereby reducing the presence of z components in the C-line's framing illustrated in Fig. 1e of the main text. Note that expressing the amplitude of the field of E_+^p relative to \mathbf{E}^k becomes an ill-defined task because the depicted fields are formed with polynomial beams [3], which are not

normalizable. We therefore abstain from relatively expressing the amplitude of E_+^p by simply expressing both fields as polynomials.

Reviewer #2: The notation of the fields should be homogenized with the supplementary note 3: e.g. the superindex k (which I assume stands for knot) is not used in the latter.

Reply: The superindex k indeed stands for knot. The notation used in the main text and Supplementary Note 3 have been homogenized in the latest iteration of the manuscript. Namely, E_-^{mp} and E_z^{mp} in Supplementary Note 3 have respectively been replaced by E_-^k and E_z^k in order to be consistent with the main text.

Reviewer #2: Does the superindex p stand for planewave? If so, why not use k since it is used to construct the knot?

Reply: The superindex p indeed stands for planewave. We reserved the k superindex for knotted polarization terms that explicitly need each other to form a knotted field that satisfies Maxwell's equations. More specifically, the knotted component E_-^k requires the presence of E_z^k in order to be a valid solution hence the use of the superindex k for both terms. Though important in creating the C-lines investigated in this work, the plane wave E_+^p is a valid solution to Maxwell's equation on its own that does not exhibit knotted features, hence we have decided to label it with a p.

Reviewer #2: While the discussion about the role of E_+^p in the knot structure is interesting, its relevance to the current work is not clear. Does the field E_+^p affect the stability and/or size of the knot? And how do you choose its value? Is it related to the Gaussian beam used in the experimental implementation?

Reply: Given that increasing the amplitude of E_+^p makes the topology of the C-line transition from that of the phase vortices of E_z^k to those of E_-^k , then there indeed exists a range of E_+^p amplitudes where the knot becomes unstable. This comment has been addressed in the following addition to Supplementary Note 3.

The sudden transition observed between the topologies of the C-lines formed with $E_+^p=0$ and $E_+^p=5$ further elucidates the importance of E_+^p in our framed knots' construction. Given that a knotted C-line formed with $|E_+^p|=0$ is entirely determined by the topology of E_z^k whereas that formed with $|E_+^p| \gg 0$ is entirely determined by the topology of E_-^k , then there exists a range of $|E_+^p|$ where the topology of the knotted C-line becomes unstable as it transitions from one form to the other. Note that such transitions are of little practical importance for structures realized within paraxial fields given the negligible presence of the z-polarized field.

The last sentence of this passage also clarifies how the amplitude of E_+^p is not relevant in our experiments given the paraxial nature of our beams and hence the negligibility of z-polarized components of our fields. This thought was also included in the last sentence of the aforementioned passage that was modified in the main text.

Note that E_z^k is negligible for paraxial beams, which are the main experimental focus of this work. Hence, for such beams, the C-line aligns with the aforementioned knotted vortices regardless of the amplitude of E_+^p [21]

In our experiments, the role of E_+^p is indeed taken over by the large Gaussian beam with the opposite circular polarization. This aspect of our work has been addressed in the following passage:

The other is modulated to form a large Gaussian beam that uniformly covers the entirety of the knotted component, thereby effectively taking the role of the plane wave E_+^p in Fig. 1c.

Reviewer #2: It is hard to see what is going on in Fig. 1c-e; they are too small. The scale used for the axes should also be indicated.

Reply: Fig. 1 has now been modified in order to make the contents of Fig. 1c-e more noticeable. Axes are now also labeled.

Reviewer #2: For the knotted lines in Fig. 1d it might be worth adding the case $E_{p+}=0$.

Reply: The case of $E_{p+}=0$ has now been added to Fig. 1d-e.

Reviewer #2: Given that most of the nonparaxial treatment is done with polynomial beams, the results shown in Fig. 6 seem like an afterthought. The generation of optical knots with focused fields using Richards-Wolf theory has already been considered in Ref. [32]. Therefore, unless the effect of focusing on the frame of the knot is discussed, there is no need for Fig. 6.

Reply: Fig. 6 and discussions related to it have been removed from the manuscript. In order to emphasize the importance of reference [32] in obtaining more interesting framed topologies, we have provided the following sentence in the revised version of our work:

Indeed, a wealth of structures, including the trefoil and cinquefoil knots investigated here, are predicted to form over a distance comparable to the optical field's wavelength [34].

Reviewer #2: Concerning the prime encoding scheme, I have some minor comments: The parameter α should be a positive *integer* and not a positive real number, right? If not, N would not be a natural number.

Reply: In principle, α can be a positive real number, upon which Alice and Bob agree beforehand, this doesn't change the main idea. However, in the context of prime factorization, which is the main application currently presented, α should indeed be a positive integer. We don't object making this change in the text and have therefore correspondingly modified passages where α is presented as a positive real number such that they now consider α as a positive integer. In the main text, these passages include

The use of this method relies on a pair of numbers (α, β) where α is a positive integer, and β is a number both related to α and to the topological structure of the framed knot.

She then proceeds by performing the following steps. She first chooses a positive integer a .

whereas in the Supplementary, they include

If we now assign a distinct prime p_k with each such strand while constraining a to positive integers, then the framed braid may be thought of as representing the prime factorization of some natural number encoded by the underlying framed knot.

Choose a positive integer a .

Reviewer #2: Once Bob receives the pair (α, β) and the framed knot, he then proceeds to measure M , the total number of half twists in the framed knot. With these three parameters and their relation to the dk parameters he can retrieve them. However, they are retrieved as an *unordered* set of integers which can be used to recover an equivalent framed braid representation. Unless I am wrong, this should be stated explicitly in the main text. Right now it is only alluded at the very end of Supplementary Note 2.

Reply: That's right. There's some arbitrariness which can be easily solved by means of an accepted convention that Alice and Bob previously adopted. We now explain this in the text by means of the following passage:

To prevent the latter from being retrieved as an unordered set of integers, Alice and Bob rely on a previously adopted convention clarifying how the extracted d_k are assigned to distinct strands of the encoded braid

Reviewer #2: Other minor comments: Freund was the first to show the existence of Mobius bands in optical fields, so his work [e.g. Opt. Commun. 283, 1–15 (2010) and Opt. Lett. 35, 148–150 (2010)] should be cited alongside Refs. [7,8].

Reply: The references mentioned by the reviewer have now been included in our work and are cited alongside what used to be references 7 and 8.

Reviewer #2: In Sup. Fig. 7 it is hard to tell that the knots become shorter with b .

Reply: We have added another viewpoint of Sup. Fig. 7 in order to make the compression more noticeable.

Reviewer #3: In the revised manuscript, the authors have substantially improved the presentation of the manuscript, particularly on the prime encoding scheme with framed knots and their experiment.

As such, they have in the process addressed all my concerns, many of which also overlaps with Reviewer #1's.

With the more substantial experiment description of among the first practical application of optical knots, I now consider their work ready for publication in Nature Communications. As far as I know, this is the first experimental work that illustrates a potential application of a type of topology different and much more sophisticated than the usual topological invariants of bandstructures i.e. Chern numbers and Z_2 indices.

Reply: We would like to thank the reviewer for their positive feedback and are glad to hear that our previous edits have addressed their comments.